# GATED NEURAL ODES: TRAINABILITY, EXPRESSIVITY AND INTERPRETABILITY

## ABSTRACT

Understanding how the dynamics in biological and artificial neural networks implement the computations required for a task is a salient open question in machine learning and neuroscience. In particular, computations requiring complex memory storage and retrieval pose significant challenge for these networks to implement or learn. Recently, a family of models described by neural ordinary differential equations (nODEs) has emerged as powerful dynamical neural network models capable of capturing complex dynamics. Here, we extend nODEs by endowing them with adaptive timescales using gating interactions. We refer to these as gated neural ODEs (gnODEs). Using a task that requires memory of continuous quantities, we demonstrate the inductive bias of the gnODEs to learn (approximate) continuous attractors. We further show how reduced-dimensional gnODEs retain their modeling power while greatly improving interpretability, even allowing explicit visualization of the structure of learned attractors. We introduce a novel measure of expressivity which probes the capacity of a neural network to generate complex trajectories. Using this measure, we explore how the phase-space dimension of the nODEs and the complexity of the function modeling the flow field contribute to expressivity. We see that a more complex function for modeling the flow field allows a lower-dimensional nODE to capture a given target dynamics. Finally, we demonstrate the benefit of gating in nODEs on several real-world tasks.

## 1 INTRODUCTION

How can the dynamical motifs exhibited by an artificial or a biological network implement certain computations required for a task? This is a long-standing question in computational neuroscience and machine learning (Vyas et al., 2020; Khona & Fiete, 2022). Recurrent neural networks (RNNs) have often been used to probe this question (Mante et al., 2013; Vyas et al., 2020; Driscoll et al., 2022), as they are flexible dynamical systems that can be easily trained (Rumelhart et al., 1986) to perform computational tasks. RNNs, particularly ones that incorporate gating interactions (Hochreiter & Schmidhuber, 1997; Cho et al., 2014), have been wildly successful in solving complex real-world tasks (Jozefowicz et al., 2015).

While RNN models provide a link between dynamics and computation, how their (typically) high-dimensional dynamics implement computation remains hard to interpret. On this note, we may turn to neural ordinary differential equations (nODEs), a class of dynamical models with a velocity field parametrized by a deep neural network (DNN), which can potentially implement more complex computations in lower dimensions than classical RNNs (Chen et al., 2018; Kidger, 2022).[1] This increased complexity in lower latent/phase-space dimensions subsequently helps in extracting interpretable, *effective* low-dimensional dynamics that may underlie a dataset or task (Kim et al., 2021).

Despite their promise, nODEs remain under-explored in the following crucial aspects. **Trainability**: Can we improve performance of nODEs by introducing gating interactions (Hochreiter & Schmidhuber, 1997; Cho et al., 2014) to tame gradients in dynamical systems? **Expressivity**: How does

---

[1] By classical RNNs, we mean the form of RNNs often considered in the neuroscience, physics and cognitive-science literature, where the interaction between units are additive, and the interaction strengths are represented by a matrix (McCulloch & Pitts, 1943; Sompolinsky et al., 1988; Elman, 1990; Vogels et al., 2005; Sussillo & Abbott, 2009; Song et al., 2016; Yang et al., 2019).

the structure of the neural network modeling the velocity flow field influence a nODE's capacity to model complex trajectories? **Interpretability**: Does the capability of low-dimensional nODEs to model complex data improve interpretability of the dynamical computation? We summarize below the main insights of our exploration of these questions.

**Main Contributions**

- We leverage our understanding of gating interactions to introduce the *gated neural ODE* (gnODE). We find that gating endows nODEs with adaptive timescales, and improves trainability of nODEs on tasks involving long timescales or rich representations (Section 2, Appendix B).

- We introduce a novel measure of expressivity related to the capacity of a neural network to store complex dynamical trajectories. nODEs and gnODEs are more expressive compared to RNNs in many parameter regimes (Sections 4, 5.3, Appendices C, F).

- We demonstrate an inductive bias of gnODEs and other gated networks to utilize marginally-stable fixed points in a "flip-flop" task that requires storing continuous memory. We further demonstrate the interpretability of the gnODEs' solutions, which organize the marginally-stable fixed-points in an approximate continuous attractor (Section 5.2, Appendix E).

- We show the advantage of gating in nODEs on real-world tasks (Sections 5.4–5.5, Appendix G).

- We determine the critical initialization for nODEs using dynamical mean-field theory (Appendix A).

## 2 GATED NEURAL ODE

The **gated neural ODE** (gnODE) is described by

$$\tau \dot{\boldsymbol{h}} = G_\varphi(\boldsymbol{h}, \boldsymbol{x}) \odot \left[ -\boldsymbol{h} + F_\theta(\boldsymbol{h}, \boldsymbol{x}) \right], \tag{1}$$

where $\tau$ is the time constant, $\boldsymbol{h} \in \mathbb{R}^N$ is the hidden/latent state vector, and $\boldsymbol{x}(t) \in \mathbb{R}^D$ is the input vector. The velocity vector field $F_\theta : \mathbb{R}^N \times \mathbb{R}^D \to \mathbb{R}^N$ and the gating function $G_\varphi : \mathbb{R}^N \times \mathbb{R}^D \to \mathbb{R}^N$ are parameterized (via $\theta$ and $\varphi$, respectively) by neural networks. While $F_\theta$ and $G_\varphi$ in general can each be parametrized by any neural network, in this work, we restrict $F_\theta$ and $G_\varphi$ to fully-connected feedforward neural networks (FNN) $F_\theta(\boldsymbol{h}, \boldsymbol{x}) = \boldsymbol{s}^{L_h}$ and $G_\varphi(\boldsymbol{h}, \boldsymbol{x}) = \boldsymbol{s}^{L_z}$, where

$$\boldsymbol{s}^1 = \phi_a(\boldsymbol{W}_*^0 \boldsymbol{h} + \boldsymbol{U}_* \boldsymbol{x} + \boldsymbol{b}_*^0), \quad \boldsymbol{s}^{\ell_*+1} = \phi_a(\boldsymbol{W}_*^{\ell_*} \boldsymbol{s}^{\ell_*} + \boldsymbol{b}_*^{\ell_*}), \quad \boldsymbol{s}^{L_*} = \phi_*(\boldsymbol{W}_*^{L_*-1} \boldsymbol{s}^{L_*-1} + \boldsymbol{b}_*^{L_*-1})$$

with $* \in \{h, z\}$. Here, $\boldsymbol{W}_*^\ell \in \mathbb{R}^{N_{\ell_*+1} \times N_{\ell_*}}$, $\boldsymbol{s}^{\ell_*} \in \mathbb{R}^{N_{\ell_*}}$, $\boldsymbol{b}_*^{\ell_*} \in \mathbb{R}^{N_{\ell_*+1}}$, and $N_0 = N_{L_*} = N$ is the phase-space (or latent) dimension. $\phi_h \in \{\mathcal{I}, \tanh\}$ and $\phi_z = \sigma$, where $\mathcal{I}$ is the identity function and $\sigma(x) = [1 + e^{-x}]^{-1}$. When $L = 1$, $\phi_a = \phi_*$. When $L > 1$, we typically set $\phi_a$ to be ReLU.

Without the leak term $-\boldsymbol{h}$ and the gating interaction (i.e., setting $G_\varphi(\boldsymbol{h}, \boldsymbol{x}) = \boldsymbol{1}$), this reverts to a form in which nODEs are typically studied (Chen et al., 2018): $\tau \dot{\boldsymbol{h}} = F_\theta(\boldsymbol{h}, \boldsymbol{x}(t))$.[2] We include the leak term $-\boldsymbol{h}$ in our formulation because it allows us to initialize the weights of the (gated or non-gated) nODE in either the stable or critical regime. Without the leak term, we show that the nODE is *always* dynamically unstable for any initialization, except for the zero initialization, and we expect this to hinder training (Abarbanel et al. (2008); see Appendix A for details).

When we set $L_h = L_z = 1$, Equation (1) reduces to a "minimal gated recurrent unit" (mGRU; Ravanelli et al. (2018)), which is a simplified version of the popularly used gated recurrent unit (GRU; Cho et al. (2014)). When in addition the gating interaction is removed ($G_\varphi(\boldsymbol{h}, \boldsymbol{x}) = \boldsymbol{1}$), Equation (1) reduces to a widely studied class of models known as "Elman" (or "vanilla") RNNs.[3]

---

[2] In Chen et al. (2018), $\tau = 1$ and $\boldsymbol{x}(t) = t$.

[3] $\tau \dot{\boldsymbol{h}} = -\boldsymbol{h} + \boldsymbol{W}_h^0 \phi_h(\boldsymbol{h}) + \boldsymbol{U}_h^0 \boldsymbol{x} + \boldsymbol{b}_h^0$ is also popular in neuroscience models, where $\boldsymbol{h}$ can be interpreted as the internal voltage of a neuron, and $\phi_h(\boldsymbol{h})$ as the output firing rate of the neuron; $W_{h,ij}^0$ is the synaptic strength between neuron $j$ and neuron $i$ (Sompolinsky et al., 1988).

Can & Krishnamurthy (2021); Krishnamurthy et al. (2022) show that the mGRU exhibits a manifold of marginally-stable fixed points in the limit of step-like gating function $\sigma$ for a wide range of parameters. This property is likely involved in shaping the inductive bias of gated networks, since it is useful in tasks requiring memory of continuous quantities (see Appendix B for an analysis of Jacobian spectrums of networks assuming different architectures, gated or non-gated).

## 3 RELATED WORK

Our work is closely related to neural controlled differential equations (nCDEs), developed in Kidger et al. (2020), which prescribes a principled way to include inputs with nODEs: $\tau \dot{\boldsymbol{h}} = \tilde{F}_\theta(\boldsymbol{h}) \frac{d\boldsymbol{x}(t)}{dt}$. An important distinction between nODE and nCDE is that the nODE takes in input $\boldsymbol{x}(t)$, whereas nCDE uses the *time-derivative* of the input $d\boldsymbol{x}/dt$. Because the choice of the interpolation scheme used in nCDE also determines how the derivative is estimated, which scheme to use becomes critical (Morrill et al., 2021). nODE avoids the complication of calculating the derivative, though it may not be as general as the nCDE (Kidger et al., 2020).

The primary motivation for introducing gating is its robust ability to generate long timescales and to address the exploding and vanishing gradients problem (EVGP) (Hochreiter & Schmidhuber, 1997; Pascanu et al., 2013; Cho et al., 2014; Can et al., 2020; Krishnamurthy et al., 2022). Our work can be viewed as distilling the key elements of gating from GRUs and LSTMs responsible for long timescales and stable gradients (cf. Krishnamurthy et al. (2022); Can et al. (2020)), and incorporating them in nCDE-inspired models. Future work could incorporate these gating interactions more directly in nCDEs, with their principled dealing of inputs and interpolation schemes.

Previous work explored improving the performance of nODEs by augmenting extra dimensions to the phase space (Dupont et al., 2019) or by regularizing nODEs to encourage simpler dynamics (Kelly et al., 2020; Ghosh et al., 2020; Finlay et al., 2020; Pal et al., 2021). Gating can be applied in addition to these improvements, which we expect will make gnODE more powerful. We also expect to see that gating will be beneficial for related model classes, such as neural stochastic differential equations (nSDEs) (Li et al., 2020).

Notable recent works have used RNNs based on discretized ODEs to deal with the EVGP, and achieve near state-of-the-art performance on various tasks. A RNN based on a system of coupled non-linear oscillators (coRNN) was introduced in Rusch & Mishra (2020), and this was extended to a Hamiltonian system with multiple (learned) time-scales in Rusch & Mishra (2021). In particular, the presence of the learned timescales was important for solving tasks with long timescales (Rusch & Mishra, 2021). In Rusch et al. (2021), the authors introduced a RNN based on gated ODEs – the long expressive memory (LEM) – that makes the timescales *adaptive* and effectively deals with the EVGP.

LEM in Rusch et al. (2021) is in fact a special case of mGRU (i.e., $L_h = L_z = 1$ in Equation (1)) where the second half of the columns of $\boldsymbol{W}_z^0$ is constrained to be zero and $\boldsymbol{W}_h^0$ is constrained to an anti-diagonal block matrix. Moreover, based on our studies of the effects of gating, we suspect that the strength of the LEM in tasks involving long memory might partially stem from gating interactions (see Appendix B for discussion). Given the strong inductive bias conferred by gating on tasks requiring long memory, our work can also be considered as extending the ODEs considered in Rusch et al. (2021) to incorporate more flexible flow-fields as in nODEs and nCDEs. We include LEM in our experiments on real-world datasets for comparison (see Sections 5.4–5.5).

Finally, in addition to addressing the EVGP, we show in this work that gating introduces a powerful inductive bias for integrator-like behavior (see Section 5.2). It achieves this by forming a continuous manifold of marginally-stable fixed points, commonly referred to as continuous attractors (defined in Appendix H; for a review, see Chaudhuri & Fiete (2016)).

## 4 EXPRESSIVITY OF A NEURAL NETWORK

In order to compare architectures, it is useful to have a principled measure of expressivity in the dynamical setting. The metric we use is inspired by the *Gardner capacity* (Gardner, 1988; Engel & Van den Broeck, 2001), which measures the ability of an architecture to interpolate a random dataset, i.e., to fit noise. The Gardner capacity is also closely linked to the VC dimension (Abbaras

et al., 2020; Engel & Van den Broeck, 2001), and was extended to temporal sequences in Bauer & Krey (1991); Taylor (1991); Bressloff & Taylor (1992).

We now introduce the relevant concepts using a discrete-time RNN of the form $\boldsymbol{h}_{t+1} = F_\theta(\boldsymbol{h}_t)$, assuming for simplicity that there is no input $\boldsymbol{x}$. The dataset we want to fit is a random time series $\xi_t = \{\xi_0, \xi_1, \xi_2, ..., \xi_T\}$. Assuming $\xi_t$ are samples from some $N$-dimensional random process, a perfect fit will require finding parameters $\theta$ which satisfy the set of $T$ equations $\xi_{t+1} = F_\theta(\xi_t)$ where $t = 0, 1, 2, ..., T-1$. The space of solutions $\theta$ at a given $T$ will occupy a region of parameter space known as the Gardner volume, which is a function of $T$. The capacity is determined by the critical sequence length $T$ at which the Gardner volume vanishes.

The longer the sequence a network can "memorize", the higher will its capacity/expressivity be. In typical systems, $T$ scales with phase-space dimension $N$ (see Appendix C for a worked-out canonical example). We suggest that an advantage of using a FNN $F_\theta$ is that the capacity instead scales with total number number of parameters, which need not scale with phase-space dimension.

Based on this notion of expressivity, in Section 5.3, we train nODEs with a variety of architectures $F_\theta$ on samples of an Ornstein-Uhlenbeck process. In our experiments, we measure instead the mean squared error between trajectories $\mathrm{MSE}(\boldsymbol{h}_t, \xi_t)$.

## 5 EXPERIMENTAL RESULTS

### 5.1 PRELIMINARIES

In our experiments, we use libraries in Julia's (Bezanson et al., 2017) SciML ecosystem, DifferentialEquations.jl and DiffEqFlux.jl (Rackauckas & Nie, 2017; Rackauckas et al., 2020), to implement all network models presented in the experiment, and choose to discretize dynamics of these networks using the canonical forward Euler method (except for the LEM, which is discretized with the forward-backward Euler method, following Rusch et al. (2021)). We use the "discretize-then-optimize" approach to obtain the gradient of the loss with respect to the network parameters (for more discussion on different choices of discretization and adjoint, see Appendix D.3 and D.4). Whenever there are missing values in a dataset, we used natural cubic splines to interpolate the missing values, following Kidger et al. (2020).

### 5.2 $N$-BIT FLIP-FLOP TASK

We examine how a vanilla RNN, mGRU, GRU, nODE and gnODE implement the "$n$-bit flip-flop task" (Sussillo & Barak, 2013). In the original $n$-bit flip-flop task (Sussillo & Barak, 2013), the network is given a continuous stream of inputs coming from $n$ independent channels. In each channel, a transient pulse of value either $+1$ or $-1$ is emitted at random times. The network should continuously generate $n$-dimensional outputs, where each dimension of the outputs should maintain the value of the most recent pulse in each channel (see Appendix E.1 for an illustration). Because each output channel of the network should take one of two values, the network should generate one of $2^n$ outputs at each time point.

Consistent with previous findings (Sussillo & Barak, 2013), when we trained our networks (see Appendix D.1 for training details) on the 3-bit flip-flop task, we find that all networks we consider can reach validation mean squared error (MSE) $< 0.01$ on the task, for a range of different phase space dimensions $N$, with appropriate hyperparameters. We also find that all networks use similar strategies to solve the task, with each of the $2^3$ stable fixed points representing each output that the networks can take (see Appendix E.1 for details).

**Variable-Amplitude Flip-Flop Task** We then modified the task so that each pulse in each channel takes a real value sampled uniformly from $-1$ to $1$ (Figure 1A). We trained our networks from one of 27 different combinations of hyperparameters (i.e., learning rates, rates of weight decay and batch sizes; see Appendix E.2 for details). When we set the phase-space dimension of our networks to be $N = 6$, we find that gnODE successfully reached validation MSE $< 0.01$ with appropriate hyperparameters, while for other networks, all runs reached MSEs $\geq 0.025$ (Figure 1B). We verified that the validation MSEs of our networks converged after training (Figure 1C). This suggests that

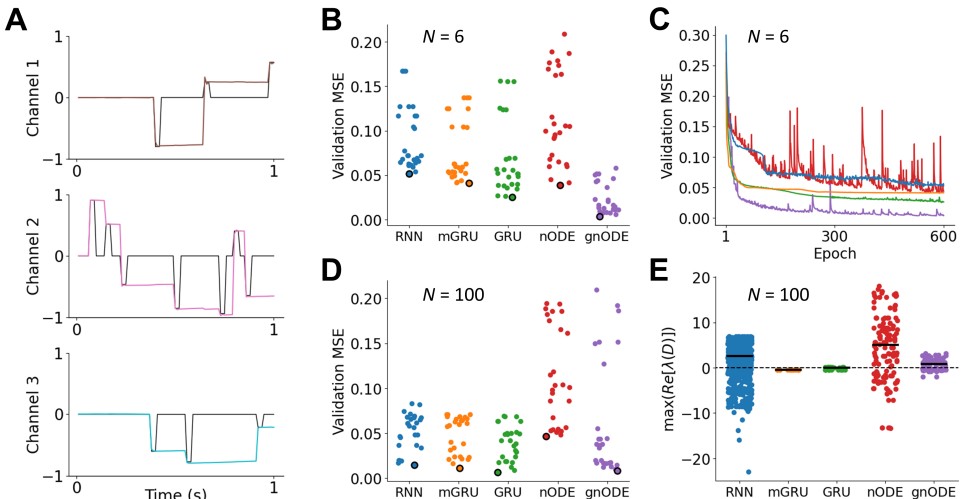

Figure 1: Networks assuming $N = 6$ (A–C) and $N = 100$ (D–E) performing the variable-amplitude 3-bit flip-flop task. (A) An example validation trial with inputs in each channel shown in black, and the trained gnODE traces maintaining the previous pulse value shown in colors. (B) For each network, we tried 27 different hyperparameter configurations. Each circle represents the minimum validation MSE achieved during 600 epochs of training. Circles with black edges represent the minimum out of the 27 configurations. (C) Validation loss traces as a function of epoch is shown for the circles with black edges. Color codes are the same as in (B). (D) Same as (B). (E) Each circle is the spectral abscissa of the Jacobian evaluated at a detected fixed point. Bold horizontal lines indicate medians.

**only gnODE is able to solve the task accurately when the phase-space dimension is low** (i.e., $N = 6$).

In contrast, when the phase-space dimension is high ($N = 100$), we find that the gnODE and GRU reached validation MSE $< 0.01$ and the vanilla RNN and mGRU reached validation MSE $< 0.016$ with appropriate hyperparameters (Figure 1D). Thus, vanilla RNN, mGRU, GRU and gnODE can solve the task in high phase-space dimensions.

**Structure of Solutions: Fixed-Points and Marginal Stability** Following Sussillo & Barak (2013), to examine how these networks solve the task, we use Newton's method initialized from points in the trajectories taken by these networks, and find solutions that reach $\|\dot{\boldsymbol{h}}\| < 0.01$ (see Appendix E.3 for details on the fixed-point finding algorithm). For each 100-dimensional network that reached the minimum validation MSE among the 27 different hyperparameter configurations, we ran $10,000$ starting points to detect fixed points, and computed the maximum real component of the eigenvalues (i.e., **spectral abscissa**) of the numerical Jacobian obtained from each of the detected fixed points. The distribution of these spectral abscissas shows that the medians and the quartiles of the gated networks (mGRU, GRU, gnODE) are closer to zero, compared to those of the vanilla networks (vanilla RNN, nODE) (Figure 1E). This suggests that we detected more (effectively) marginally-stable fixed points for the gated networks compared to the vanilla networks. For the vanilla networks, we see that many of the detected fixed points are stable (i.e., spectral abscissas are much less than zero), in contrast to the gated networks. This suggests that a vanilla RNN may be reaching its solution using a combination of marginally-stable and stable fixed points, while the gated networks mostly rely on marginally-stable fixed points to reach their solutions. We obtained similar results for networks assuming $N = 6$, although for these networks, only gnODE reached validation MSE $< 0.01$ (see Appendix E.4 for details).

**Interpretability of gnODEs** While analyses on the 100-dimensional vanilla RNN, mGRU, GRU and gnODE trained on the task can give useful insights, we found that when we apply PCA on the trajectories taken by these networks, we needed at least 10 principal components to reach more than 0.9 variance explained, suggesting that the high-dimensional networks do not necessarily favor low-

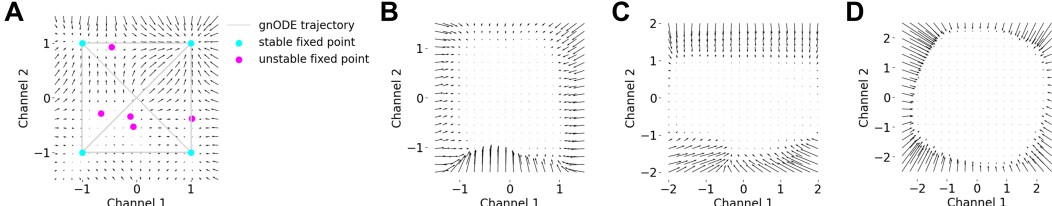

Figure 2: Flow fields of gnODE with $N = 2$ performing four different versions of the 2-bit flip-flop task. The input pulses $c_1$ in Channel 1 and $c_2$ in Channel 2 are given by (A) $c_1, c_2 \in \{-1, +1\}$. The light gray lines indicate example trajectories taken by gnODE. (B) **Square**: real-valued input pulses in the interval $c_1, c_2 \in [-1, 1]$. Numerically-identified fixed points are omitted for better visualization. (C) **Rectangle**: $c_1 \in [-2, 2]$ and $c_2 \in [-1, 1]$. (D) **Annulus**: $1 \le \sqrt{c_1^2 + c_2^2} \le 2$.

dimensional solutions in this setup. However, in principle, a dynamical system as simple as the one taking up 3 dimensions, which has a cube filled with marginally-stable fixed points, can solve this task. Indeed, we find that when we set the phase-space dimension of gnODE to be $N = 3$, it can still achieve validation MSE $< 0.01$ with appropriate hyperparameters. We were not able to achieve this low MSE for other networks, suggesting gnODEs might be appropriate for studying the emergence of *interpretable* solutions to the variable-amplitude flip-flop task.

For simplicity, we turned to training a 2-dimensional gnODE on the 2-bit flip-flop task and its variants and plot the 2 dimensional flow field such that the two axes describing this space are projected onto the axes that correspond to the outputs, Channel 1 and Channel 2. For the fixed-amplitude task (where the pulse values can either be $+1$ or $-1$), we find 4 stable fixed points, and find that each input perturbation moves the gnODE state from exactly one stable fixed point to another (Figure 2A). We then trained a gnODE on a variable-amplitude 2-bit flip-flop task where the pulses can take values from $-1$ to $1$. When we plot the flow field in the output space, we see that the velocity of the flows are close to zero, and this plane of fixed points roughly form a square between $-1$ and $1$. Input perturbations try to move the gnODE state within the square, so that gnODE can hold onto the memory of the inputs (Figure 2B). In summary, the gnODE learns a **continuous attractor** in the shape of a square, and is solving the variable-amplitude flip-flop task in an intuitively appealing way, by simply integrating the input.

The plane of fixed points show up not only for this particular task but also for other tasks. Instead of varying the values of the pulses from $-1$ to $1$, we varied the values of the pulses in Channel 1 from $-2$ to $2$ and find a **rectangular attractor** (Figure 2C). We also tried varying the statistics of the pulses so that pulses in the two channels are no longer independent, but appear at the same time, and the value taken by the pulse in Channel 1, $c_1$, and the value taken by the pulse in Channel 2, $c_2$, satisfy $1 \le \sqrt{c_1^2 + c_2^2} \le 2$. We see a **disk attractor** with radius roughly of 2 in this case. We do not see a hole between radius 0 and 1 because crossing this region may be the fastest way from one state to another, and we did not explicitly penalize the network for crossing this region (Figure 2D). Consistent with the flow field, we find that, even though the gnODE has not seen any inputs with pulse values satisfying $0 < \sqrt{c_1^2 + c_2^2} < 1$ during training, when it is given inputs with pulse values satisfying $0.5 < \sqrt{c_1^2 + c_2^2} < 1$, it generalizes well (MSE $= 0.005$; see Appendix E.5 for details). However, when it is given inputs with pulse values that are small (i.e., $0 < \sqrt{c_1^2 + c_2^2} < 0.5$), it does tend to mistake them as having no input at all, resulting in worse performance (MSE $= 0.028$). When the gnODE is given inputs with pulse values $2 < \sqrt{c_1^2 + c_2^2} < 4$, it does not generalize (MSE $= 0.490$).

We do not plot the flow fields for other networks assuming $N = 2$, as all of the 27 runs with different hyperparameter configurations reached validation MSEs $> 0.05$ for vanilla RNN, mGRU and GRU, and validation MSEs $> 0.02$ for nODE (see Appendix E.5 for details).

These results together suggest that gnODEs might be flexible enough to learn more general manifold geometries, provided they are trained on an appropriate synthetic task. Furthermore, the geometry of the low-dimensional representations found by gnODEs can directly inform their generalization capacities, thanks to their enhanced interpretability (see Figure 8 in Appendix E.5 for flow fields of gnODE trained on more variants of the 2-bit flip-flop task).

## 5.3 FITTING AN ORNSTEIN-UHLENBECK TRAJECTORY

We introduce a task to measure the practical expressivity of a neural network. The task that the network has to perform is to perfectly fit a finite number of samples from an Ornstein-Uhlenbeck (OU) process,

$$\tau_{OU} d\boldsymbol{z} = \lambda_{OU} \boldsymbol{z} dt + \boldsymbol{x} dt + \sigma_{OU} d\boldsymbol{w},$$

where $\boldsymbol{w}$ is a Wiener process. As long as $\tau_{OU}$ is sufficiently smaller than $S/d$ where $S$ is the total length of the trajectory and $d$ is the distance between consecutive samples, we have samples that are reasonably uncorrelated. In our analysis, we set $\dim(\boldsymbol{z}) = 30$, $\tau_{OU} = 1$s, $\lambda_{OU} = -1$, $\boldsymbol{x}(t) = \boldsymbol{1}$, and $\sigma_{OU} = 1$, and sample at every 1s of this trajectory for 100s (therefore having a total of 100 samples). We train our networks on a single trajectory of these 100 samples. For the vanilla RNN, mGRU and GRU, we systematically vary the phase-space dimension $N$ and $\tau$ of the model. For the nODE and gnODE, along with $N$ and $\tau$, we also vary the number of hidden layers in $F_\theta$ and the number of units $N_\ell$ in each hidden layer of $F_\theta$.

We generally see that, for all networks, when the model $\tau$ is closer to $\tau_{OU}$, we achieve lower training MSEs, confirming our intuition that networks perform best when their timescales match correlation time of the data (Figure 3A–B). We also confirmed that generally when we increase the number of units $N_\ell$ in each hidden layer, the networks become more expressive. Figure 3C shows an example of this for gnODEs assuming $\tau_{OU} = \tau = 1$s, and 1 hidden layer in $F_\theta$ (see Appendix F for results with nODEs and for different numbers of layers). The other side of this same coin is that hidden layers can act as a bottleneck for expressivity. We can see this in Figure 3C, where for large phase-space dimension, a small hidden layer can hurt expressivity. We also see that for various regimes, gnODE can be more expressive than other networks especially when $N$ is low (Figure 3E–F; see Appendix F for analyses not highlighted in the main text).

By changing the model $\tau$ on a given dataset, we are effectively changing the difficulty of the task that the networks have to solve. Transients in the network will be relevant on timescales that scale as $\sim \tau$; therefore, for very large $\tau$, the velocity is suppressed and $\boldsymbol{h}$ evolves very slowly. This places a greater burden on $F_\theta$ to send small changes in the phase space into effectively orthogonal vectors in the OU time series. Therefore, we suspected that in the transient regime, the complexity of $F_\theta$ becomes more important for fitting noise. Confirming our intuition, we see that as $\tau$ increases, the performance gap between networks that have more complex $F_\theta$ (i.e., nODEs and gnODEs) and networks with simpler $F_\theta$ (i.e., RNNs, mGRUs and GRUs) becomes larger (Figure 3D–F).

## 5.4 REAL-WORLD TASKS

### 5.4.1 LATIN ALPHABET CHARACTER TRAJECTORY CLASSIFICATION

In this task, networks of different architectures were trained to classify 20 different Latin alphabet characters from irregularly-sampled time series consisting of the $x$ and $y$ positions of the pen tip and the force on the tip. This dataset ("CharacterTrajectories") is originally from the UEA time series classification archive (Bagnall et al., 2018), and we used the preprocessed data obtained from the Neural CDE repository[4] (see Appendix G.2 and Kidger et al. (2020) for details). We trained each network by performing a grid search over the hyperparameter space to find the set of hyperparameters that minimizes the validation loss (see Appendix G.2 for details). Table 1A shows the accuracy of each network on the classification task.

### 5.4.2 WALKER2D KINEMATIC SIMULATION PREDICTION

The networks were given the task of predicting the dynamical evolution of the trajectories generated by the MuJoCo physics engine kinematic simulations (Todorov et al., 2012). The preprocessed data for this task were obtained from the ODE-LSTM repository,[5] (see Appendix G.3 and Lechner & Hasani (2020) for details). While Lechner & Hasani (2020); Xia et al. (2021) did not choose to interpolate missing data with natural cubic splines, doing so helps with performances of the networks as we show in Table 1B – we generally see MSEs that are lower than those reported in

---

[4]`https://github.com/patrick-kidger/NeuralCDE`
[5]`https://github.com/mlech26l/ode-lstms`

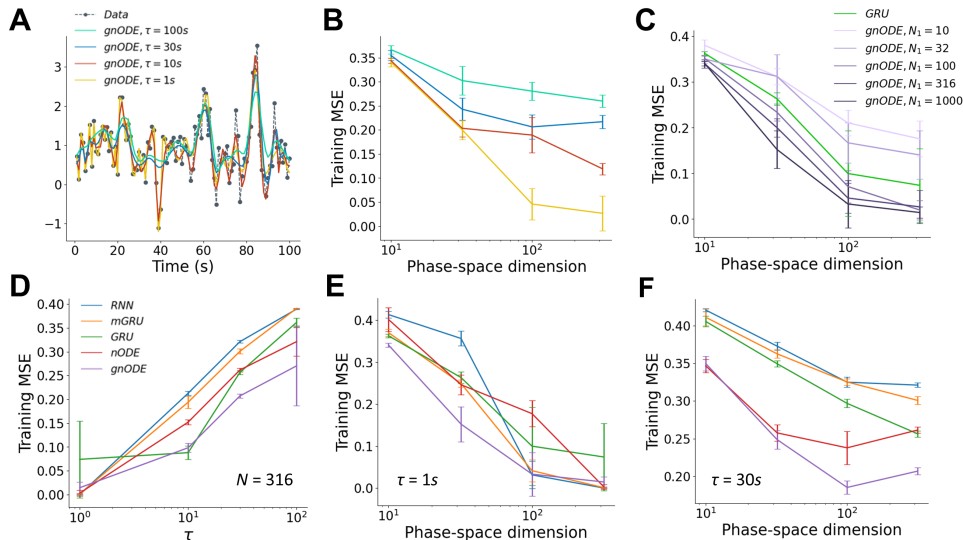

Figure 3: Practical expressivities of vanilla RNN, mGRU, GRU and gnODE, and their dependence on network timescale. (B–F) show means and standard deviations across 5 runs. (A) gnODE with 1 hidden layer ($N = N_1 = 316$) assuming $\tau \in \{1s, 10s, 30s, 100s\}$ fitting samples from the OU trajectory. (B) Training MSEs of gnODE in (A). (C) Training MSEs of gnODE with 1 hidden layer, assuming $\tau = 1s$. (D) $N = 316$ across all networks, $N_1 = 1000$ for nODE and gnODE. (E) $\tau = 1s$ across all networks, $N_1 = 1000$ for nODE and gnODE. (F) $\tau = 30s$ across all networks, $N_1 = 1000$ for nODE and gnODE.

Lechner & Hasani (2020); Xia et al. (2021) (the lowest reported MSE on this task is $0.883 \pm 0.014$, with an ODE-LSTM). Table 1B shows the test MSE of each network on the prediction task, with the hyperparameters that achieved the lowest MSE on the validation dataset (see Appendix G.3 for details).

### 5.4.3 SPEECH COMMANDS CLASSIFICATION

We trained the networks on the task of classifying ten spoken words, such as "Stop" and "Go", based on audio recordings of these words. The dataset is originally from Warden (2018) and preprocessed using the pipeline in the Neural CDE repository (see Appendix G.4 and Kidger et al. (2020) for details). Table 1C shows the test accuracy of each network on the classification task, with the hyperparameters that achieved the highest accuracy on the validation dataset (see Appendix G.4 for details).

## 5.5 CONCLUSIONS FROM THE REAL-WORLD TASKS

We observe in Table 1 that, when the phase-space dimension $N$ is fixed to be 100 for all networks, the gnODE performs competitively against other architectures considered in all of the three real-world tasks (Sections 5.4.1–5.4.3). We also performed hyperparameter-search over different $N$s and generally find that gnODEs with lower $N$s can perform competitively against other networks with higher $N$s (see Appendices G.2–G.5 for more details and discussion on the fairness of comparisons). This suggests a practical advantage in choosing gnODE over other networks for tasks involving sequences.

Consistent with results in Section 5.3, we show in Appendix G.4 how much the intrinsic time constant $\tau$ can influence the results of training. While nODE performs almost at chance level for the $\tau$ considered in Table 1C, we find that changing $\tau$ can significantly improve nODE performance. We also find that gnODE appears most robust to changes in $\tau$, suggesting that it would be a reasonable default architecture when one is lacking a principled approach to choosing the time constant (Appendix G.4).

We additionally show that, for some datasets, the critical initialization determined in Appendix A, when used together with $F_\theta(\boldsymbol{h}, \boldsymbol{x})$ that has tanh as the final nonlinearity, can enhance performance of a gnODE on those datasets (Appendices G.2–G.4). Furthermore, we show that a gating function $G_\varphi(\boldsymbol{h}, \boldsymbol{x})$ with $L_z > 1$ can give a boost in performance compared to a gating function $G_\varphi(\boldsymbol{h}, \boldsymbol{x})$ with $L_z = 1$ in gnODEs (Appendix G.4).

Table 1: Networks assuming $N = 100$ performing real-world tasks. (A) Test accuracy (mean $\pm$ std, across 5 runs) on CharacterTrajectories. (B) Test MSE on Walker2D kinematic simulation. (C) Test MSE on SpeechCommands.

| | (A) CharacterTrajectories | (B) Walker2D | (C) SpeechCommands |
|---|---|---|---|
| Model | Test Accuracy | Test MSE | Test Accuracy |
| mGRU | $0.983 \pm 0.002$ | $1.138 \pm 0.030$ | $0.809 \pm 0.018$ |
| GRU | $0.988 \pm 0.003$ | $0.850 \pm 0.032$ | $0.819 \pm 0.004$ |
| LSTM | $0.987 \pm 0.002$ | $0.865 \pm 0.009$ | $0.768 \pm 0.012$ |
| LEM | $0.988 \pm 0.003$ | $0.709 \pm 0.009$ | $0.794 \pm 0.007$ |
| nODE | $0.898 \pm 0.089$ | $0.707 \pm 0.023$ | $0.140 \pm 0.012$ |
| gnODE | $0.986 \pm 0.003$ | $0.588 \pm 0.003$ | $0.823 \pm 0.006$ |

## 6 DISCUSSION

We introduced gated neural ordinary differential equations (gnODEs), a novel nODE architecture which utilizes a gating interaction to dynamically and adaptively modulate the timescale. A synthetic $n$-bit flip-flop task (cf. Sussillo & Barak (2013)) was used to demonstrate the inductive bias of the gnODEs to learn continuous attractors. We also showed that, compared to other architectures, the gnODE can learn this task with a lower phase-space dimension. This allows us to inspect the nature of the solution learned in an intuitive and interpretable manner. We also formulated a principled measure of expressivity for RNNs/nODEs based on their ability to fit random trajectories. We used this measure to investigate how the phase-space dimension and the complexity of the velocity field interact to shape the overall expressivity. We saw that when the phase-space dimension is low, the gnODE can be more expressive compared to the other architectures tested. Lastly, even though gating results in more parameters and slower per-iteration update of the network state, we empirically showed that a gated network (whether it be a gated RNN or a gated nODE) can significantly improve performance compared to a vanilla network, both on carefully designed synthetic tasks and real-world tasks.

While we do not claim that each unit in a gnODE can correspond to a biological neuron, there is evidence that biological neural networks utilize several of the mechanisms that are found in the gnODE. First, gating appears to be a generally observed phenomenon in biological neural networks. For example, a gnODE can, similar to an LEM, be mapped onto a network of Hodgkin-Huxley neurons where gating corresponds to voltage-gated ion channels (Rusch et al., 2021). In another example, negative-derivative feedback in an E-I balanced network can be viewed as a form of gating which dynamically changes the time constant (Lim & Goldman, 2013). Furthermore, it is known that a gating mechanism allows a network to robustly form continuous attractors (Can & Krishnamurthy, 2021), which is thought to be prevalent in biological neural networks (Khona & Fiete, 2022). Second, recent experiments show that neural population activities across a large number of brain regions and species can be described by a low-dimensional dynamical system (Churchland et al., 2012; Harvey et al., 2012; Mante et al., 2013; Kaufman et al., 2014; Nieh et al., 2021). However, our work shows that high-dimensional networks do not necessarily favor a low-dimensional solution to a low-dimensional task.

Among the networks that we considered in this work, gnODE is the only network that both uses a gating mechanism and is capable of learning complex dynamics even in low phase-space dimensions, consistent with the previous literature on how biological neural networks work. These features make gnODE a uniquely powerful model for probing the connection between computation and dynamics in artificial and biological neural networks.

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

## A  CRITICAL INITIALIZATION FOR NEURAL ODEs

In this Appendix, we will determine the critical initialization for neural ODEs. First, defining the model

$$\dot{\boldsymbol{h}} = -\boldsymbol{h} + F_\theta(\boldsymbol{h}, \boldsymbol{x}) \tag{2}$$

We will consider a multi-layer perceptron (MLP) network function $F_\theta$ defined according to the equations

$$F_\theta(\boldsymbol{h}, \boldsymbol{x}) \stackrel{\text{def}}{=} \boldsymbol{a}^L \tag{3}$$

$$\boldsymbol{a}^{\ell+1} = \boldsymbol{W}^\ell \phi(\boldsymbol{a}^\ell) + \boldsymbol{b}^\ell, \quad \text{for} \quad \ell = 1, ..., L-1, \tag{4}$$

$$\boldsymbol{a}^1 = \boldsymbol{W}^0 \boldsymbol{h} + \boldsymbol{U}\boldsymbol{x} + \boldsymbol{b}^0, \tag{5}$$

$$\boldsymbol{W}^\ell \in \mathbb{R}^{N_{\ell+1} \times N_\ell}, \quad \boldsymbol{b}^\ell \in \mathbb{R}^{N_{\ell+1}}, \quad \boldsymbol{a}^\ell \in \mathbb{R}^{N_\ell}, \quad \boldsymbol{h} \in \mathbb{R}^N, \quad \boldsymbol{x} \in \mathbb{R}^D. \tag{6}$$

This is equivalent to the feedforward neural networks (FNN) defined in the main text under the identification $\boldsymbol{s}^L = \boldsymbol{a}^L$ and $\boldsymbol{s}^\ell = \phi(\boldsymbol{a}^\ell)$ for $\ell = 1, ..., L-1$. We have also separated $\boldsymbol{W}^0$ from $\boldsymbol{U}$, because they should be scaled differently.

**Jacobian**  A useful quantity in studying the dynamics and assessing stability is the instantaneous Jacobian $\mathcal{D}$. This will be related to the input-output Jacobian $\mathcal{J}$ of the MLP, where

$$\mathcal{J}_{ij} = \frac{\partial (F_\theta)_i}{\partial h_j} = \left( W^{L-1}[\phi'(\boldsymbol{a}^{L-1})]\boldsymbol{W}^{L-2}[\phi'(\boldsymbol{a}^{L-2})]....\boldsymbol{W}^1[\phi'(\boldsymbol{a}^1)]\boldsymbol{W}^0 \right)_{ij}. \tag{7}$$

Using this, the instantaneous Jacobian of the nODE is

$$\mathcal{D}_{ij} = -\delta_{ij} + \mathcal{J}_{ij}. \tag{8}$$

### A.1  MEAN-FIELD THEORY

**Initialization and Mean-Field Scaling**  We consider two choices of scaling which lead to a mean-field theory, each informed by popular initialization schemes in machine learning. The first is the Kaiming scaling of the weights:

$$W_{ij}^\ell \sim \mathcal{N}\left(0, \frac{\sigma_w^2}{N_\ell}\right), \quad \text{Kaiming scaling} \tag{9}$$

with $N_0 = N_L = N$ being the dimension of the phase space in which $\boldsymbol{h}$ lives. We also naturally would like $U_{ij} \sim \mathcal{N}(0, \sigma_u^2/D)$, in order for the input to not be unnecessarily suppressed by $N_0$. This is only a problem if $N$ and $D$ are significantly mismatched.

Alternatively, we can take inspiration from the popular Glorot initialization and use

$$W_{ij}^\ell \sim \mathcal{N}\left(0, \frac{\sigma_w^2}{N_\ell + N_{\ell+1}}\right), \quad \text{Glorot scaling} \tag{10}$$

The mean-field theory then requires taking $N_\ell \to \infty$ (including $N_0$) while keeping their ratios fixed.

Defining the aspect ratio

$$\alpha_{\ell+1} = N_{\ell+1}/N_\ell, \tag{11}$$

we will develop the results below assuming the following initialization scheme

$$W_{ij}^\ell \sim \mathcal{N}\left(0, \frac{\sigma_\ell^2}{N_\ell}\right), \quad \sigma_\ell^2 = \sigma_w^2/(1 + \alpha_{\ell+1}) \tag{12}$$

Keeping $\alpha_\ell$ makes this equivalent to Glorot scaling, whereas setting all $\alpha_\ell = 0$ recovers Kaiming scaling.

By keeping $\sigma_w$ unspecified, we have actually introduced more flexibility to what is typically understood by these initialization schemes. In fact, what is usually called Kaiming/Glorot initialization has $\sigma_w = \sqrt{2}$. We will keep to this convention, and refer to Kaiming/Glorot *scaling* when $\sigma_w$ is not explicitly fixed.

**Correlation Functions in MFT**   The dynamical mean-field theory (DMFT) for the nODE follows the logic presented in many previous works, see e.g. Crisanti & Sompolinsky (2018); Helias & Dahmen (2020). Proceeding via the Martin-Siggia-Rose statistical field theory, in the saddle-point approximation, valid for large $N$, $\boldsymbol{h}$ is described by a Gaussian process with zero mean and covariance determined by the self-consistent DMFT equation

$$(\partial_t + 1)(\partial_{t'} + 1)C_h(t, t') = C_F(t, t'), \tag{13}$$

where we have chosen the convention to represent correlation functions

$$C_h(t, t') = \langle \frac{1}{N} \sum_i h_i(t) h_i(t') \rangle_\theta, \quad C_F(t, t') = \langle \frac{1}{N} \sum_i F_i(t) F_i(t') \rangle_\theta, \tag{14}$$

and the averages are taken over the random parameters.

In order to find a self-consistent solution, we need to express $C_F$ as a function of $C_h$. This can be accomplished by appealing to well-known known results in the literature on the neural network Gaussian process (NNGP) kernel for the MLP defined by $F_\theta$ (see e.g. Williams (1996); Lee et al. (2017)). To get the desired correlation function, or kernel, we define a hidden layer kernel function

$$K^\ell(t, t') = \left\langle \frac{1}{N_\ell} \sum_{i=1}^{N_\ell} a_i^\ell(t) a_i^\ell(t') \right\rangle, \tag{15}$$

which satisfies the recurrence relation

$$K^1(t, t') = \frac{\sigma_w^2}{1 + \alpha_1} C_h(t, t') + \sigma_u^2 C_x(t, t') + \sigma_b^2, \tag{16}$$

$$K^{\ell+1}(t, t') = \frac{\sigma_w^2}{1 + \alpha_{\ell+1}} C_\phi(\hat{K}^\ell(t, t')) + \sigma_b^2, \tag{17}$$

$$K^{\ell+1}(t, t) = \frac{\sigma_w^2}{1 + \alpha_{\ell+1}} C_\phi(K^\ell(t, t)) + \sigma_b^2, \tag{18}$$

$$C_F(t, t') = K^L(t, t'). \tag{19}$$

where

$$\hat{K}^\ell(t, t') = \begin{pmatrix} K^\ell(t, t) & K^\ell(t, t') \\ K^\ell(t', t) & K^\ell(t', t') \end{pmatrix}. \tag{20}$$

Here, we have defined the correlators

$$C_\psi(\hat{K}) = \int \frac{d^2\mathbf{x}}{2\pi \det \hat{K}} e^{-\frac{1}{2}\mathbf{x}^T \hat{K}^{-1} \mathbf{x}} \psi(x_1) \psi(x_2) \tag{21}$$

$$C_\psi(K) = \int \frac{dx}{2\pi K} e^{-\frac{x^2}{2K}} \psi(x)^2. \tag{22}$$

**Asympotic Stability**   Let us consider the divergence of trajectories. The usual trick is to take two replicas with different initial conditions but identical weights (Derrida & Pomeau, 1986; Schuecker et al., 2018). This will change the DMFT in the following way

$$(\partial_t + 1)(\partial_{t'} + 1)C_h^{ab}(t, t') = C_F^{ab}(t, t'), \tag{23}$$

with $a, b = 1, 2$. Here, the RHS is obtained from the recurrence relations

$$K^{1,ab}(t, t') = \frac{\sigma_w^2}{1 + \alpha_1} C_h^{ab}(t, t') + \sigma_b^2, \tag{24}$$

$$K^{\ell+1,ab}(t, t') = \frac{\sigma_w^2}{1 + \alpha_{\ell+1}} C_\phi \left[ \hat{K}^{\ell,ab}(t, t') \right] + \sigma_b^2, \tag{25}$$

$$C_F^{ab}(t, t') = K^{L,ab}(t, t'). \tag{26}$$

We assume a steady state which is time-translation invariance, so the correlation functions depend only on the difference $\tau = |t - t'|$. Then, expanding around the replica symmetric solution $C_h^{12}(\tau) = C_h(\tau) + \epsilon Q(\tau) e^{\lambda T}$ will give the eigenvalue equation for $Q$

$$\left((\lambda + 1)^2 - \partial_\tau^2\right) Q = \chi_L(\tau) Q, \tag{27}$$

where we have used

$$\frac{\partial C_F}{\partial C_h^{12}(\tau)} = \frac{\sigma_w^2}{1 + \alpha_L} C_{\phi'}(\hat{K}^{L-1}) \dots \frac{\sigma_w^2}{1 + \alpha_2} C_{\phi'}(\hat{K}^1) \times \frac{\sigma_w^2}{1 + \alpha_1} \tag{28}$$

$$= \chi_L(\tau). \tag{29}$$

Here, we have defined the susceptibility $\chi_\ell(t, t')$ which satisfies its own recurrence relation (suppressing the time arguments)

$$\chi_{\ell+1} = \frac{\sigma_w^2}{1 + \alpha_{\ell+1}} C_{\phi'}\left(\hat{K}^\ell\right) \chi_\ell, \quad \chi_1 = \frac{\sigma_w^2}{1 + \alpha_1}, \tag{30}$$

The susceptibility at unequal times is typically not studied in the FNN setting (Schoenholz et al., 2017; Doshi et al., 2021). Usually, the equal-time susceptibility $\chi(0)$ is sufficient, since it characterizes the behavior of gradients. The object which appears here $\chi(\tau)$ is tantamount to studying the overlaps of the gradient of the FNN output for two different inputs. However, if we are instead interested in fixed points, we have quite simply

$$(\lambda + 1)^2 = \chi_L(0). \tag{31}$$

The susceptibility which appears here $\chi(0)$ is precisely the object typically studied for FNN. So, if we use the intuition from feedforward networks and initialize at criticality, we will find a marginally stable fixed point in the nODE.

**Fixed-Point Jacobian Radius**  Proceeding, we wish to determine the edge of stability for fixed-points. To do so, we must first use the MFT to find fixed points according to the self-consistent equation

$$C_h = K^L. \tag{32}$$

In the large $N$ limit, the spectral of the Jacobian $\mathcal{D}$ depends only on the distribution of $\boldsymbol{h}$, and thus on $C_h$. Furthermore, since it is uniformly shifted by the identity, the spectral radius of $\mathcal{J}$, which we denote $\rho(\mathcal{J})$, is enough to determine stability. One can show that the squared spectral radius $\rho(\mathcal{J})$ is given by

$$\rho(\mathcal{J})^2 = \left\langle \frac{1}{N} \text{tr} \mathcal{J}^T \mathcal{J} \right\rangle \tag{33}$$

$$= \sigma_w^{2L} \prod_{\ell=1}^{L} \frac{1}{1 + \alpha_\ell} \left(\prod_{\ell=1}^{L-1} C_{\phi'}(K^\ell)\right) = \chi_L(0). \tag{34}$$

Since the correlation functions that appear depend only on the distribution of $\boldsymbol{h}$, and thus only on $C_h$, once the MFT fixed-point equation is solved, the solution can be plugged into this expression for the spectral radius to determine stability.

Note also that the squared spectral radius is equal to the static susceptibility defined above, as it must. A common set up will have $N_0 = N_L = N$, while all hidden layers have the same dimension $N_1 = \dots = N_{L-1} = H$. Then defining $\alpha = (H/N, 0)$ and $\beta = (1, 0)$ for (Glorot, Kaiming), we get

$$\rho(\mathcal{J})^2 = \frac{\sigma_w^{2L}}{(1 + \beta)^{L-2}(1 + \alpha)^2} \left(\prod_{\ell=1}^{L-1} C_{\phi'}(K^\ell)\right). \tag{35}$$

In Figure 4 we compute the critical curve in the $\sigma_w - \sigma_b$ plane along which $\rho(\mathcal{J}) = 1$. We show how this curve changes with increasing depth. For concreteness, we choose Kaiming scaling and $\phi(x) = \tanh(x)$ activation.

With biases exactly zero, the zero fixed point typically determines the edge of chaos. The spectral radius for the zero FP is

$$\rho_0^2 = \frac{\sigma_w^{2L}\phi'(0)^{2(L-1)}}{(1+\beta)^{L-2}(1+\alpha)^2}. \tag{36}$$

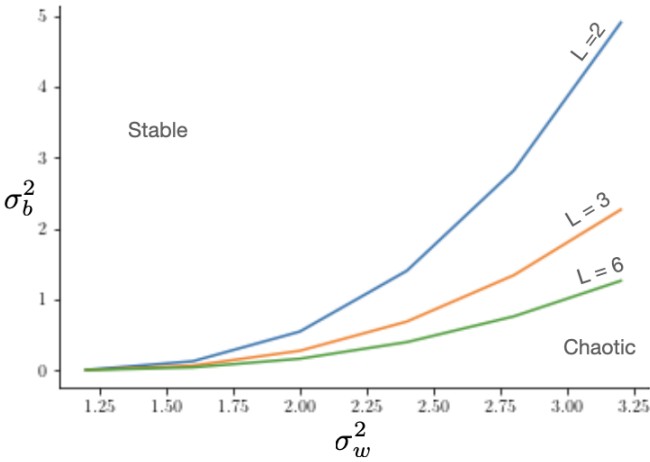

Figure 4: To obtain these curves, we used $\phi(x) = \tanh(x)$. This shows the critical curve separating stability from chaos as a function of bias and weight variances. The different curves correspond to MLP functions of differing depth. We used Kaiming scaling such that $\alpha = \beta = 0$. For a fixed depth $L$, the region below the plotted curve is chaotic, whereas the region above the plotted curve is stable.

**Explicit Solutions for ReLU Networks** ($\phi = \text{ReLU}$)  If the MLP utilizes only the ReLU activation, there does not appear to be a chaotic phase. When tanh is applied as the final nonlinearity for $F_\theta$, the system has a chaotic regime. The suggested initializations in Equations (51, 52) are valid for both $F_\theta$ with and without the final nonlinearity tanh.

We will make use of the integral identities for one-point functions

$$C_\phi(K) = \int Dx \left([\sqrt{K}x]_+\right)^2 = \frac{1}{2}K, \quad C_{\phi'}(K) = \int_0^\infty \frac{dh}{\sqrt{2\pi}}e^{-\frac{h^2}{2K}} = \frac{1}{2}. \tag{37}$$

and for two-point functions, setting $\mathbf{x} = (x_1, x_2)$, and assuming a time-translation invariant kernel

$$\hat{K} = \left( \begin{array}{cc} K_0 & K_\tau \\ K_\tau & K_0 \end{array} \right), \quad K_\tau \le K_0 \tag{38}$$

we have

$$C_\phi(\hat{K}) = \int_0^\infty \int_0^\infty \frac{dx_1 dx_2}{2\pi\sqrt{\det\hat{K}}} x_1 x_2 e^{-\frac{1}{2}\mathbf{x}^T \hat{K}^{-1}\mathbf{x}}, \tag{39}$$

$$= \frac{1}{4}K_\tau \left(1 + \frac{2}{\pi}\tan^{-1}\left(\frac{K_\tau}{\sqrt{K_0^2 - K_\tau^2}}\right)\right) + \frac{1}{2\pi}\sqrt{K_0^2 - K_\tau^2} \tag{40}$$

$$C_{\phi'}(\hat{K}) = \int_0^\infty \int_0^\infty \frac{dx_1 dx_2}{2\pi\sqrt{\det\hat{K}}} e^{-\frac{1}{2}\mathbf{x}^T \hat{K}^{-1}\mathbf{x}}, = \frac{1}{4}\left(1 + \frac{2}{\pi}\tan^{-1}\left(\frac{K_\tau}{\sqrt{K_0^2 - K_\tau^2}}\right)\right) \tag{41}$$

$$\tag{42}$$

**Fixed-Points**  We begin by analyzing the time-independent fixed points. The fixed-point can be determined exactly using the recurrence relations. Define the coefficients

$$a_1 = \frac{\sigma_w^2}{2(1+\alpha)}, \quad a_2 = \frac{\sigma_w^2}{2(1+\beta)}, \quad b = \sigma_b^2. \tag{43}$$

Then we can compute the kernel for the ReLU MLP via the recurrence relations

$$K^1 = 2a_1 C_h + \sigma_u^2 C_x \tag{44}$$

$$K^{\ell+1} = a_2 K^\ell + b, \quad \ell = 1, 2, ..., L - 2 \tag{45}$$

$$K^L = \frac{\sigma_w^2}{1+\alpha} \frac{1}{2} K^{L-1} + b \tag{46}$$

$$= a_1 a_2^{L-2} \left( 2a_1 C_h + \sigma_u^2 C_x \right) + a_1 \frac{1 - a_2^{L-2}}{1 - a_2} b + b. \tag{47}$$

The dynamical fixed-point of the nODE is determined by $C_h = K^L$ which implies

$$C_h = \frac{1}{1 - 2a_1^2 a_2^{L-2}} \left[ a_1 a_2^{L-2} \sigma_u^2 C_x + a_1 \frac{1 - a_2^{L-2}}{1 - a_2} b + b \right]. \tag{48}$$

Therefore, a fixed point exists for

$$2a_1^2 a_2^{L-2} < 1. \tag{49}$$

Note that since the LHS here is precisely equal to the squared spectral radius, if a fixed point exists, then it must also be stable.

Criticality will correspond to the spectral radius of the input-output Jacobian being precisely equal to unity. The resulting equation can be solved for $\sigma_w^*$ and yields

$$\sigma_w^* = (1+\alpha)^{1/L} \sqrt{2^{1-1/L}(1+\beta)^{1-2/L}}, \quad \text{Critical init.} \tag{50}$$

Specifying for the two popular initialization schemes discussed above gives

$$\sigma_w^* = \sqrt{2^{1-1/L}}, \quad \text{Kaiming scaling} \tag{51}$$

$$\sigma_w^* = 2^{1-3/2L} \left( 1 + \alpha \right)^{1/L}, \quad \text{Glorot scaling} \tag{52}$$

Comparing these to the traditional choices for these initializations, we find that Kaiming initialization with $\sigma_w^* = \sqrt{2}$ will place the network in the unstable regime. Conversely, Glorot initialization with $\sigma_w^* = \sqrt{2}$ will initialize the network in the stable regime.

A trivial corollary of our analysis thus far is that a randomly initialized nODE without a leak term is *always unstable*, since the condition for stability in this setting is $\rho(\mathcal{J}) = 0$, which implies a critical $\sigma_w^* = 0$.

## B    COMMON FEATURES OF GATING ACROSS ARCHITECTURES

Following Krishnamurthy et al. (2022), we did an analysis on the empirical Jacobian spectrum of LEM (Rusch et al., 2021) with gating and without gating, and compared them to those of mGRU ($L_h = 1, L_z = 1$) and gnODE ($L_h = 3, L_z = 1$) (Figure 5). To generate the plots in Figure 5, we set $\phi_a = \tanh$, initialized $W_{ij}^\ell$ according to Equation (9) and similarly initialized $W_{z,ij}^0$ with:

$$W_{z,ij}^0 \sim \mathcal{N}\left( 0, \frac{\sigma_z^2}{N} \right) \tag{53}$$

where $\tau = 1$s, and $N = 1000$. We discretized the network dynamics with the forward Euler method with $\Delta t = 1s$ for mGRU and gnODE, and discretized LEM with the forward-backward Euler method with $\Delta t = 1s$ (following exactly Equation (3) in Rusch et al. (2021)). To ensure that the dynamics reached steady-state, we ran the solvers up until 1000s, and evaluated the eigenvalues of the numerical Jacobian of the approximate steady-state. We found that the spectrums of the networks we get are roughly similar when we discretize the dynamics with the Tsitouras 5/4 Runge-Kutta method, except for the spectrums of the LEM, which had shapes similar to those of the mGRU.

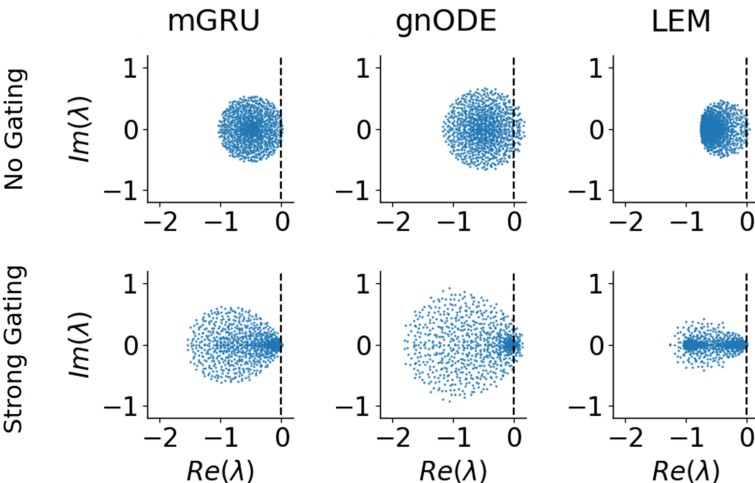

Figure 5: All non-gating weights ($\boldsymbol{W}_h^\ell$) set to Kaiming normal with overall scale $\sigma_w = 1.5$. All gating weights ($\boldsymbol{W}_z^0$) set to Kaiming normal with overall scale $\sigma_z = 0$ (i.e., no gating, top row) and $\sigma_z = 5$ (bottom row).

When the LEM does not have gating (Figure 5, top right), we see that, compared to a mGRU or a gnODE without gating (Figure 5, top left and middle), the special anti-diagonal block structure of $\boldsymbol{W}_h^0$ lets the LEM stay close to criticality. This may partially be due to the fact that the LEM without gates can be mapped to a Hamiltonian dynamical system. However, when we add gating to LEM (as presented in Rusch et al. (2021); Figure 5, bottom right), it nullifies the effect from the special anti-diagonal block structure, and we see a robust "pinching" of the Jacobian spectrum leading to eigenvalues clustering near zero and thus long timescales/stable gradients, which is ubiquitous for the gated networks (gnODE, mGRU, LEM; Figure 5, bottom row). This pinching results in long-lived modes, contributing to all of these gated networks' ability to learn long time dependencies.

## C GARDNER VOLUME FOR TRAJECTORY FITTING CAPACITY

In this section, we derive the capacity of a spherical perceptron to store a random time series by mapping the problem to Gardner's original calculation (Gardner, 1988). This result also appears in Bauer & Krey (1991); Taylor (1991); Bressloff & Taylor (1992), which studied storage capacity for time-delay RNNs. Previous work has also studied storage capacity for temporal sequences in RNNs with Hebb rule structured connectivity (Sompolinsky & Kanter, 1986; Nadal, 1988).

We start by setting up the problem in more generality. In the main text, we pursued a definition of expressivity that involved fitting a random time series. The ability to fit such noise is intimately connected to storage capacity of a perceptron.

Consider a discrete-time nODE (or a generalized RNN)

$$\boldsymbol{h}_{t+1} = F_\theta(\boldsymbol{h}_t), \tag{54}$$

with which we want to fit a random time series

$$\xi_t = \{\xi_0, \xi_1, \xi_2, ..., \xi_T\} \tag{55}$$

where $\xi_t$ are i.i.d. random variables. A perfect fit will require a set of parameters $\theta$ that satisfy the set of $T$ equations

$$\xi_{t+1} = F_\theta(\xi_t), \quad t = 0, 1, 2, ..., T-1 \tag{56}$$

We will now try to find the volume in parameter space which can satisfy this equation. A similar question was asked in Brunel (2016), which was interested in the structure of solutions which store the optimal length sequence.

We allow for an error $\epsilon$ in the fit, and we want to find all $\theta$ which satisfy these constraints. There are different formulations depending on the activation functions. In general, for smooth activation functions, we can define an indicator function

$$\chi(\boldsymbol{\xi}) = \prod_{t=0}^{T} \Theta\left(\xi_{t+1} - F_\theta(\xi_t) + \epsilon\right) \Theta\left(-\xi_{t+1} + F_\theta(\xi_t) + \epsilon\right). \tag{57}$$

If the weights are such that the trajectory of the nODE follows $\xi_t$ within some margin $\epsilon$, then $\chi = 1$; otherwise, $\chi = 0$. It is also necessary to insert some sort of regularizer, so that the volume in $\theta$ space does not explode. This will have the effect of replacing the measure $d\theta \to d\mu_\theta$ with a regularized measure that converges, and which we assume is normalized $\int d\mu_\theta = 1$. With these ingredients, the volume in the space of parameters is given by

$$V = \int d\mu_\theta \, \chi(\boldsymbol{\xi}). \tag{58}$$

Specifying this setup to the spherical perceptron considered by Gardner, we use $F(\xi) = \mathrm{sign}\left(N^{-1/2} J\xi\right)$, with parameters $J$, and binary patterns $\xi_i^t \in \{-1, +1\}$. This is the set-up analyzed in Bauer & Krey (1991); Taylor (1991); Bressloff & Taylor (1992), where it was also demonstrated that the calculation ends up being identical to the Gardner calculation. For convenience, we show here how the temporal sequence storage problem can be mapped to the storage of fixed-point storage.

Due to the threshold activation, the indicator function can be written

$$\chi(\boldsymbol{\xi}) = \prod_{t=0}^{T-1} \Theta\left(N^{-1/2} \sum_{i,j} \xi_i^{t+1} J_{ij} \xi_j^t - \epsilon\right). \tag{59}$$

The total volume will be given by

$$V = \frac{1}{Z} \int \prod_i d\mu_i \chi(\boldsymbol{\xi}), \quad d\mu_i = \prod_{j|j\neq i} dJ_{ij}\delta\left(\sum_{j|j\neq i} J_{ij}^2 - N\right), \quad Z = \int \prod_i d\mu_i. \tag{60}$$

After expressing the Heaviside step function using its Fourier representation, the expression for the volume can be seen to factorize into a product

$$V = \prod_{i=1}^{N} V_i, \tag{61}$$

where the volume $V_i$ is calculated over all entries in a fixed row $i$ of the connectivity matrix $J$:

$$V_i = \frac{1}{\int d\mu_i} \int d\mu_i \int \prod_{t=0}^{T-1} dx_t d\lambda_t \exp\left(ix_t\left(\lambda_t - N^{-1/2} \sum_{j|j\neq i} \xi_i^{t+1} J_{ij} \xi_j^t + \epsilon\right)\right), \tag{62}$$

In order to calculate the disorder (pattern) average of $\log V_i$, it is necessary to introduce replicas and calculate $\langle V_i^n \rangle$ and subsequently take $n \to 0$. The replicated volume is written

$$V_i^n = \prod_{a=1}^{n} \frac{1}{Z_i^n} \int d\mu_i^a dx_t^a d\lambda_t^a \exp\left(ix_t^a\left(\lambda_t^a - N^{-1/2} \sum_{j|j\neq i} \xi_i^{t+1} J_{ij}^a \xi_j^t + \epsilon\right)\right) \tag{63}$$

Averaging over random patterns will introduce into the integral the term proportional to

$$\prod_{t=0}^{T-1} \prod_{j|j\neq i} \cos\left(N^{-1/2} \sum_a x_t^a J_{ij}^a\right) \tag{64}$$

This is the point where we can make the mapping directly onto Gardner's calculation. Notice that after disorder averaging, the integrand factorizes into a product of terms at different times. This is identical to the factorization for different fixed-point patterns in Gardner (1988). This demonstrates that the equivalence between the volumes for fixed-point storage and temporal sequence storage is non-perturbative, and valid for any $N$. Technically, Taylor (1991); Bressloff & Taylor (1992) demonstrate the equivalence in the large $N$ setting. Thus, the calculation proceeds as in the original work, but with the the total trajectory length $T + 1$ replacing the number of patterns $p$. This yields the critical capacity as $\alpha_c = T/N = 2$. In other words, the maximal length of a trajectory scales as $T \sim 2N$.

## D  EXPERIMENT DETAILS

### D.1  CODE

All of the networks presented in this work (vanilla RNN, mGRU, GRU, LSTM, LEM, nODE and gnODE) are implemented with our Julia (Bezanson et al., 2017) package, RNNTools.jl. This package is based on Flux.jl, DifferentialEquations.jl and DiffEqFlux.jl (Innes, 2018; Rackauckas & Nie, 2017; Rackauckas et al., 2020).

### D.2  GATING ARCHITECTURE

We let the gating function $G_\varphi(\boldsymbol{h}, \boldsymbol{x})$ to be $\sigma(\boldsymbol{W}_z^0 \boldsymbol{h} + \boldsymbol{U}_z \boldsymbol{x} + \boldsymbol{b}_z^0)$ (i.e., $L_z = 1$) for Sections 5.2–5.4.2 in the main text. We default to this architecture unless otherwise noted. For Section 5.4.3, we assumed that $L_z = 2$. Anecdotally, architectures with $L_z = 2$ appears to perform better than architectures with $L_z = 1$.

### D.3  CHOICE OF DISCRETIZATION

In our experiments, we choose to discretize our networks (vanilla RNN, mGRU, GRU, LSTM, nODE and gnODE) using the canonical forward Euler method, and the LEM with the forward-backward Euler method in Rusch et al. (2021) (we also present results for LEM discretized with the forward Euler method in the corresponding Sections in the Appendix). While the optimal choice of discretization method may depend on the problem, we find that the simple Euler solver can achieve strong performance while taking less training time than an adaptive solver in our experiments. Often, the number of function evaluations (NFEs) in a nODE can become extremely large during training for adaptive schemes, and several regularization methods have been introduced to reduce NFEs (Kelly et al., 2020; Ghosh et al., 2020; Finlay et al., 2020; Pal et al., 2021). On the other hand, we can control the NFEs explicitly by changing the timestep $\Delta t$ in a fixed-timestep solver, such as the Euler method. While the Euler method does not have guarantees on the growth of error, it may in fact allow representing more functions compared to adaptive methods that provide such guarantees, precisely because of the errors from the discretization (Dupont et al., 2019). We do not lose the benefit of being able to train nODEs on irregularly-sampled time series when we use the Euler solver. For the $n$-bit flip-flop task in Section 5.2, changing the Euler method (used for presenting results in the main text) to the Tsitouras 5/4 Runge-Kutta method did not make a significant qualitative difference. For fitting our networks to the OU trajectory in Section 5.3, having an explicit control over the NFEs is crucial for a fair comparison, and the Euler solver was the natural choice. We also see that Euler discretization was sufficient to achieve good performances on the tasks in Section 5.4.1 and Section 5.4.2, which involve irregularly-sampled trajectories. For Section 5.4.1, it is interesting to see that our Euler-discretized mGRU and GRU show accuracies that are higher than the accuracies of GRU-ODEs (De Brouwer et al., 2019; Jordan et al., 2021) (which use a modern adaptive solver) reported in Kidger et al. (2020). This suggests that the Euler discretization (which does not necessarily assume $\tau = \Delta t = 1$) can be a fast, practical alternative to adaptive methods.

### D.4  CHOICE OF ADJOINT

For all networks we consider, we backpropagate through the operations of the solver—that is, we use the "discretize-then-optimize" approach, as is standard in training a RNN, instead of using the "optimize-then-discretize" approach used in Chen et al. (2018) to train nODEs. A few studies show

that the former produces more accurate gradients than the latter and can yield better performances (Gholami et al., 2019; Onken & Ruthotto, 2020).

# E $N$-BIT FLIP-FLOP TASK

For all versions of the $n$-bit flip-flop task in this section, the total length of each trial was 1s, binned into 10ms bins. Thus each trial had 100 time-bins. The width of each pulse was set to be 20ms. 600 trials were generated total, where 500 trials were used for training and the remaining 100 trials were used for validation.

Networks considered in this section (vanilla RNN, mGRU, GRU, nODE and gnODE) were initialized with Glorot uniform initialization (Glorot & Bengio, 2010) with zero bias. $\tau = 0.01$s in all of the networks. We used AdamW (Loshchilov & Hutter, 2019) for training.

## E.1 FIXED-AMPLITUDE 3-BIT FLIP-FLOP TASK

This is the version of the task that was originally introduced in Sussillo & Barak (2013). For this task, we determined the total number of pulses (summed across $n$ channels) on each trial by sampling a number $k$ from the Poisson distribution with mean 12. We then randomly chose $k$ indices from 1 to 100 without replacement. These $k$ indices were the indices at which the pulses occur. For each of the $k$ indices, we randomly chose which one of the $n$ channels the pulse will occur. Then for the channel where the pulse appears, we chose either $+1$ or $-1$ randomly as the value to be taken by the pulse (Figure 6A).

We trained our networks on 500 trials of this task for 200 epochs. The initial states of the networks were not learned, and were initialized with $\boldsymbol{h}_0 \sim \mathcal{N}(\boldsymbol{0}, \boldsymbol{\Sigma})$, where $\boldsymbol{\Sigma} = \frac{2}{N+1}\boldsymbol{I}$ was the variance. For vanilla RNN, mGRU, GRU, we varied the phase-space dimension $N = \{6, 12, 18\}$, where for each $N$, we used the learning rate $\eta = 10^{-2}$, rate of weight decay $\lambda_w = 10^{-1}$ and the batch size $B = 100$. For nODE and gnODE, we similarly varied $N = \{6, 12, 18\}$, and used $\eta = 10^{-3}$, $w = 10^{-1}$ and $B = 100$. For nODE and gnODE, $F_\theta$ had 3 hidden layers with 100 units each layer (i.e., $L = 4$ and $H = 100$). We logged the validation MSE traces of mGRU, GRU and gnODE of $N = 6$ over 200 epochs (or $200 \times (500/100) = 1000$ iterations), and found that mGRU, GRU and gnODE all achieved validation MSEs $< 0.01$ at least at some point over the 200 epochs. Similarly, we logged the validation MSE traces of vanilla RNN and nODE of $N = 6$. These networks reached minimum validation MSEs of 0.033 and 0.016, respectively, over the 200 epochs. All networks reached minimum validation MSEs $< 0.01$ during 200 epochs when $N = \{12, 18\}$ (Figure 6B). For further analyses of the trained networks (e.g., performing PCA over the trajectories taken by the networks, and finding the fixed points of the networks), we used the set of parameters that achieved the minimum validation MSEs over the 200 epochs. All networks, when the reached minimum validation MSE was $< 0.01$, used similar strategies for this task – the networks created 8 stable fixed points to solve the task, where each of the 8 stable fixed points represented each output that the networks should take (Figure 6C for vanilla RNN; other networks not shown). For details on how the fixed points were found, see Section E.3.

## E.2 VARIABLE-AMPLITUDE 3-BIT FLIP-FLOP TASK

We determined when the pulses occur and in what channel the pulses occur in the same way as Section E.1. Then for the channel where the pulse appears, we drew a sample $m$ from $U[-1, 1]$ and let $m$ be the value to be taken by the pulse (Figure 1A in main text).

To ensure fair comparisons across different networks (vanilla RNN, mGRU, GRU, nODE and gnODE), for each network, we ran $3 \times 3 \times 3 = 27$ different configurations of $(\eta, w, B)$, where $\eta = \{10^{-4}, 10^{-3}, 10^{-2}\}$, $w = \{10^{-3}, 10^{-2}, 10^{-1}\}$ and $B = \{10, 50, 100\}$. For each network and each configuration, we trained for 600 epochs, and determined the set of parameters that gives the minimum validation MSE over the 600 epochs. Each circle in Figure 1B is the minimum validation MSE achieved over 600 epochs for a single configuration, with a total of 27 circles for each network. For nODE and gnODE, $F_\theta$ had 3 hidden layers with 100 units each layer (i.e., $L = 4$ and $H = 100$). We let $N$ be either 6 (Figure 1A–C) or 100 (Figure 1D–E).

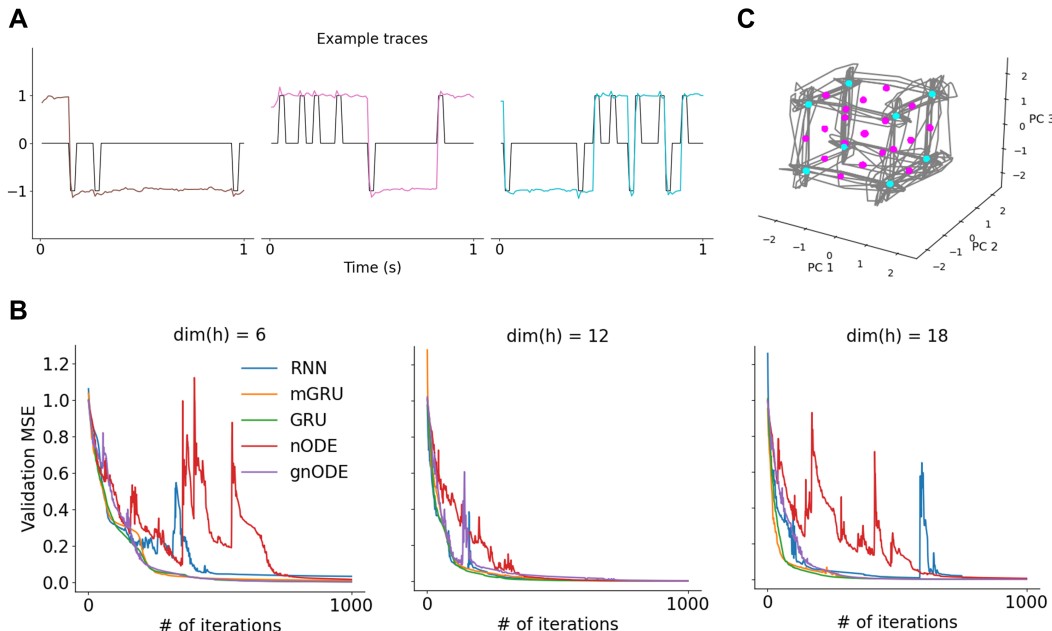

Figure 6: Networks performing the original fixed-amplitude 3-bit flip-flop task (Sussillo & Barak, 2013). (A) An example validation trial with inputs in each channel shown in black, and the trained vanilla RNN traces maintaining the previous pulse value shown in colors. (B) Validation loss traces as a function of the number of iterations. (C) The first 3 principal components of the vanilla RNN ($N = 18$) trajectories and fixed points. Cyan indicates stable fixed point. Magenta indicates unstable fixed point.

### E.3 FIXED-POINT FINDER

The finder should find some $h$ which satisfies $\dot{h} \approx 0$. To find such $h$, we define some function $f$ such that $\dot{h} = f(h)$. In the case of a nODE, for example, $f(h) = (1/\tau) \cdot \left(-h + F_{\hat{\theta}}(h, x(t))\right)$, where we assume that $x(t) = 0$, and $\hat{\theta}$ is the set of trained parameters of the nODE. We used Newton's method (implemented in Julia's NLsolve.jl package; Mogensen & Riseth (2018)) to find the root of the nonlinear function $f(h)$. From a starting point, we ran the method for 100 iterations, and terminated whenever $\|f(h)\| < 0.01$. Choosing what starting point to use can be important, especially when $N$ is large. Following Sussillo & Barak (2013), we used points in the trajectories taken by the network in the validation trials as the starting points of the finder. We detected fixed points by running the finder $10,000$ times, each with $10,000$ different starting points. Once we detect some $h$ that satisfies $\|\dot{h}\| < 0.01$, we checked whether each element $h_i$ for all $i \in \{1, 2, ..., N\}$ satisfy $2q < h_i < 2r$ to ensure that the detected fixed (or slow) point is not too far from the trajectories taken by the network. Here, $q = \min_y (\min_i(y_i))$ where $y$ is one of the points in the trajectories taken by the network in the validation trials. Similarly, $r = \max_y (\max_i(y_i))$.

We also explored a different criterion to identify fixed points that are near the latent trajectories – whenever the identified fixed point is less than 1 in Euclidean distance from any of the points actually traversed by the networks, we include the fixed point in the plot. Even for this criterion, we still saw a result similar to what was presented in Figure 1E.

### E.4 STABILITY OF FIXED POINTS

Figure 1E suggests that the vanilla RNN ($N = 100$) may be reaching its solution using a combination of marginally-stable and stable fixed points, while the gated networks (mGRU, GRU and gnODE) mostly rely on marginally-stable fixed points.

We further projected the 100-dimensional fixed points of the vanilla RNN to the 3-dimensional PC space and found that the unstable fixed points are scattered around the stable fixed points, suggest-

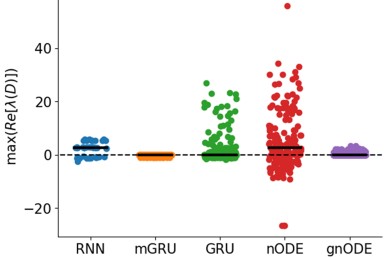

Figure 7: Networks assuming $N = 6$ performing the variable-amplitude 3-bit flip-flop task. Each circle is the spectral abscissa of the Jacobian evaluated at a detected fixed point. Bold horizontal lines indicate medians.

ing that the unstable fixed points may be facilitating the network to fall into one of the stable or marginally stable fixed points.

We did a similar analysis for networks assuming $N = 6$ and find similar results (Figure 7). The medians and quartiles of the plotted circles in Figure 1E of the main text and those of Figure 7 are provided in Table 2.

Table 2: We compute the spectral abscissa $\max(\mathrm{Re}\,[\lambda(\mathcal{D})])$ at a numerically-detected fixed point. We provide below the medians and quartiles of the distribution of $\max(\mathrm{Re}\,[\lambda(\mathcal{D})])$.

(a) $N = 100$

| Model | Median | Quartiles |
|-------|--------|-----------|
| RNN | 2.587 | $[-1.969, 6.360]$ |
| mGRU | $-0.472$ | $[-0.473, -0.470]$ |
| GRU | $-0.033$ | $[-0.474, 0.070]$ |
| nODE | 4.988 | $[-1.627, 9.453]$ |
| gnODE | 0.842 | $[-0.461, 1.698]$ |

(b) $N = 6$

| Model | Median | Quartiles |
|-------|--------|-----------|
| RNN | 2.750 | $[-0.768, 4.703]$ |
| mGRU | 0.004 | $[0.002, 0.009]$ |
| GRU | 0.005 | $[0.001, 0.023]$ |
| nODE | 2.626 | $[-2.030, 13.151]$ |
| gnODE | 0.002 | $[0.002, 0.003]$ |

### E.5 THE FAMILY OF 2-BIT FLIP-FLOP TASKS

#### E.5.1 4 STABLE FIXED POINTS

We determined when the pulses occur and in what channel the pulses occur in the same way as Section E.1, except that now $n = 2$. We trained the gnODE for 200 epochs with 27 different hyperparameter configurations, similar to Section E.2. The initial state of the gnODE was learned – the initial state was assumed to be an affine transformation of the input at the first time-bin. The gnODE's $F_\theta$ had 2 hidden layers with 316 units each layer (i.e., $L = 3$ and $H = 316$). For Figure 2A, we used the gnODE that reached the lowest validation MSE (3.203e-8) among the 27 different runs.

#### E.5.2 SQUARE ATTRACTOR

We determined when the pulses occur and in what channel the pulses occur in the same way as Section E.2, except that now $n = 2$ (Figure 8A). We trained all networks (vanilla RNN, mGRU, GRU, nODE and gnODE) for 200 epochs, each with 27 different hyperparameter configurations, similar to Section E.2 (Figure 8B). The initial states of the networks were learned – the initial state was assumed to be an affine transformation of the input at the first time-bin. For nODE and gnODE, $F_\theta$ had 2 hidden layers with 316 units each layer (i.e., $L = 3$ and $H = 316$). For Figure 2B, we used the gnODE that reached the lowest validation MSE (0.008) among the 27 different configurations.

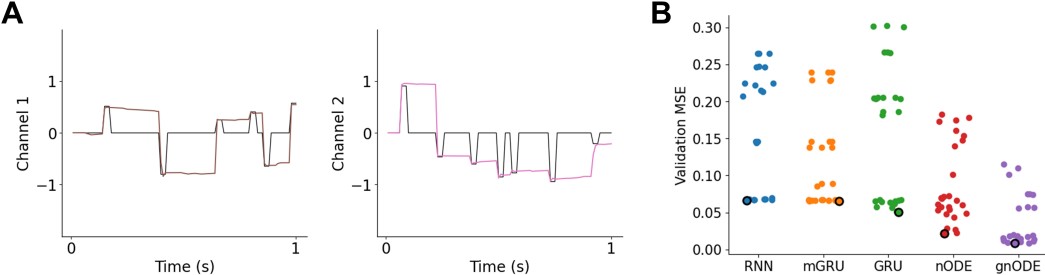

Figure 8: Networks assuming $N = 2$ performing the square 2-bit flip-flop task (Section E.5.2). (A) An example validation trial with inputs in each channel shown in black, and the trained gn-ODE traces maintaining the previous pulse value shown in colors. (B) For each network, we tried 27 different hyperparameter configurations. Each circle represents the minimum validation MSE achieved during 200 epochs of training. Circles with black edges represent the minimum out of the 27 configurations.

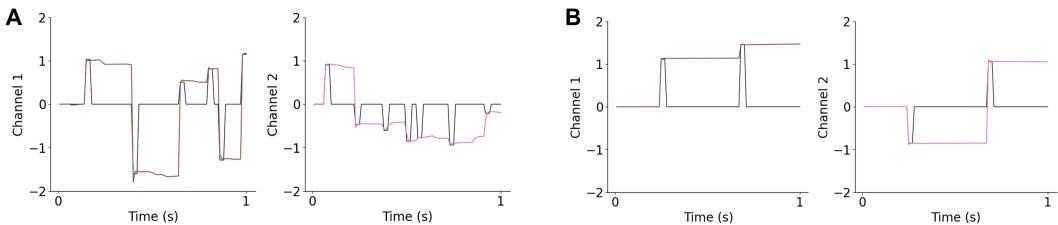

Figure 9: The gnODE assuming $N = 2$ performing the rectangle and the disk 2-bit flip-flop task. (A) An example validation trial on the rectangle task with inputs in each channel shown in black, and the trained gnODE traces maintaining the previous pulse value shown in colors. (B) Same as (A) but for the disk task.

### E.5.3 RECTANGLE ATTRACTOR

We determined when the pulses occur and in what channel the pulses occur in the same way as Section E.5.2, except that the pulse value for Channel 1 was drawn from $U[-2, 2]$, while the pulse value for Channel 2 was drawn from $U[-1, 1]$ (Figure 9A). For Figure 2C, we used the gnODE that reached the lowest validation MSE (0.0137) among the 27 different configurations. Architecture used for gnODE for this task was the same as the one used in Section E.5.2.

### E.5.4 DISK ATTRACTOR

We determined the total number of pulses (summed across $n$ channels) on each trial by sampling a number $k$ from the Poisson distribution with mean 6. We then randomly chose $k$ indices from 1 to 100 without replacement. These $k$ indices were the indices at which the pulses occur. For each of the $k$ indices, we drew random samples $c_1, c_2, ..., c_n$ which satisfy $1 < \sqrt{c_1^2 + c_2^2 + ... + c_n^2} < 2$, where $c_i$ is the pulse value in Channel $i$. Thus the input pulses in $n$ channels appear at the same time. For Figure 2D, we used the gnODE that reached the lowest validation MSE (1.910e-4) among the 27 different configurations. Architecture used for gnODE for this task was the same as the one used in Section E.5.2.

### E.5.5 RING ATTRACTOR

We determined when the pulses occur in a way similar to Section E.5.5. However, $k$ was drawn from the Poisson distribution with mean 12, not 6. Also, the constraint that $c_1, c_2, ..., c_n$ had to satisfy was $\sqrt{c_1^2 + c_2^2 + ... + c_n^2} = 2$.

We trained the gnODE for 200 epochs with 27 different hyperparameter configurations on this task. Architecture used for gnODE for this task was the same as the one used in Section E.5.2. Figure 10 shows the flow fields of gnODE trained on this task. Figure 10A shows the flow field of gnODE

when we let the initial state that the network should take (in the output space) be $(0, 0)$ (validation MSE: 8.345e-5). We see a structure that is similar to what we see when we train the gnODE on the disk 2-bit flip-flop task (Section E.5.4). When we instead let the initial state that the network should take be $(2, 0)$, we see a more ring-like structure near $(2, 0)$ (Figure 10B; validation MSE: 0.007). We also tried changing the initial state of gnODE from $(0, 0)$ to $(1.5, 0)$ for the gnODE trained on the disk 2-bit flip-flop task (Section E.5.4), but did not see a significant qualitative difference.

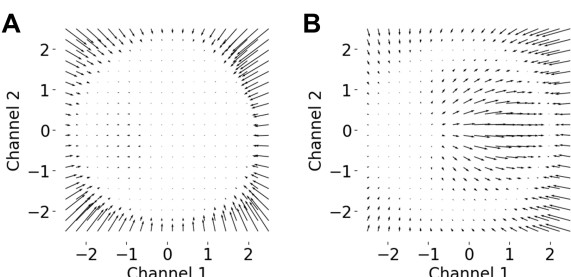

Figure 10: Flow fields of gnODE with $N = 2$ performing the ring 2-bit flip-flop task. (A) The initial state of gnODE is set to be $(0, 0)$. (B) The initial state of gnODE is set to be $(2, 0)$.

### E.6 How are Continuous Attractors Generated in these tasks?

When we evaluated the gate output $G_\varphi(\boldsymbol{h}, \boldsymbol{x})$ (with $L_z = 1$) and $-\boldsymbol{h} + F_\theta(\boldsymbol{h}, \boldsymbol{x})$ on several points inside the attractor, the norm of $-\boldsymbol{h} + F_\theta(\boldsymbol{h}, \boldsymbol{x})$ was generally closer to 0 than the norm of the gate output. Thus, after training, it may be possible for gnODE to do the tasks without the gate. However, this does not mean that gates do not help with training/performance. For the points evaluated, we found that each component of the gate had values $\sim 0.1$, and this helps $-\boldsymbol{h} + F_\theta(\boldsymbol{h}, \boldsymbol{x})$ become even closer to zero.

## F Fitting an Ornstein-Uhlenbeck Trajectory

The task that the network has to perform is to perfectly fit a finite number of samples from an Ornstein-Uhlenbeck (OU) process,

$$\tau_{OU} d\boldsymbol{z} = \lambda_{OU} \boldsymbol{z} dt + \boldsymbol{x} dt + \sigma_{OU} d\boldsymbol{w}, \tag{65}$$

where $\boldsymbol{w}$ is a Wiener process. In our analysis, we set $\dim(\boldsymbol{z}) = 30$, $\tau_{OU} = 1\text{s}$, $\lambda_{OU} = -1$, $\boldsymbol{x}(t) = \boldsymbol{1}$, and $\sigma_{OU} = 1$, and generate a single trajectory from this process for 100s using the SOSRI method[6] and sample at every 1s of this trajectory. This gives us a total of 100 samples from this OU process. We trained our networks on a single trajectory of these 100 samples in a single batch, using AdamW (Loshchilov & Hutter, 2019). We trained vanilla RNN, mGRU, GRU, nODE and gnODE on this task. The network weights were initialized with Glorot Normal initialization (Glorot & Bengio, 2010), and biases were initialized with a zero-mean Gaussian with variance $10^{-6}$. The networks received $\boldsymbol{x}(t) = \boldsymbol{1}$ as their inputs. Note that $\dim(\boldsymbol{x}) = \dim(\boldsymbol{z}) = 30$. The initial states of the networks were learned – the initial state was assumed to be an affine transformation of the input at the first time-bin. For the vanilla RNN, mGRU and GRU, we systematically varied the phase-space dimension $N$ and $\tau$ of the model. For the nODE and gnODE, we set $L_z = 1$, and varied the number of hidden layers $L = L_h$ in $F_\theta$ and the number of units $N_\ell$ in each hidden layer of $F_\theta$, along with $N$ and $\tau$. We assumed that the number of units $N_\ell$ is the same across the hidden layers (i.e., $H = N_1 = ... = N_{L-1}$).

To ensure we are using the appropriate learning rate and the rate of weight decay, we trained each network with 9 different combinations of the learning rate and the rate of weight decay, and picked the best model out of the 9 that reached the lowest training loss anytime during the 2000 epochs of training. The training MSE values plotted in Figure 3 of the main text consider only the lowest training losses out of the 9. With the learning rates and rates of weight decay determined, we ran

---

[6]This is the default SDE solver in the DifferentialEquations.jl package in Julia (Rackauckas & Nie, 2017).

the experiment 5 times with different random seeds. We chose the learning rate from one of $10^{-4}$, $10^{-3}$ and $10^{-2}$, and the rate of weight decay from $10^{-3}$, $10^{-2}$ and $10^{-1}$.

We see that for all networks, when the model $\tau$ is closer to $\tau_{OU} = 1$, we generally achieve lower training MSEs (Figure 3D and Figure 11). This confirms our intuition that networks perform best when their timescales match correlation time of the data.

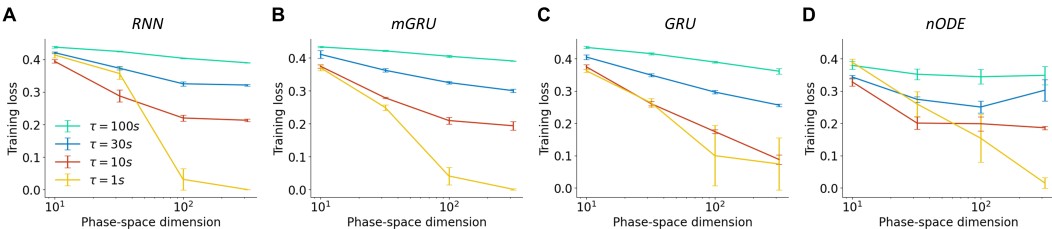

Figure 11: (A) Vanilla RNN assuming $\tau \in \{1s, 10s, 30s, 100s\}$ fitting samples from the OU trajectory. (B) mGRU. (C) GRU. (D) nODE with 1 hidden layer ($H = N_1 = 316$).

We also see that, for nODE and gnODE, when we increase the number of units $H$ in each hidden layer, the networks become more expressive (Figure 3C and Figure 12). When model $\tau = 1s$, nODEs tend to be more expressive than vanilla RNNs when the phase-space dimension is low, but we see that as we increase the phase-space dimension, RNNs become more expressive (Figure 12A–B). However, as we increase model $\tau$, nODEs become consistently more expressive than RNNs when $H$ is sufficiently large (Figure 12C–D and Figure 13A–C). We find that gnODEs are consistently more expressive than GRUs across different model $\tau$'s ($\tau \in \{1s, 10s, 30s, 100s\}$) and different numbers of phase-space dimensions $N$ (Figure 3C and Figure 12E and Figure 13A–C). This confirms our intuition that increasing $\tau$ is equivalent to effectively increasing the difficulty of the task that the networks have to solve, and that, because $h$ evolves very slowly for very large $\tau$, this places a greater burden on $F_\theta$.

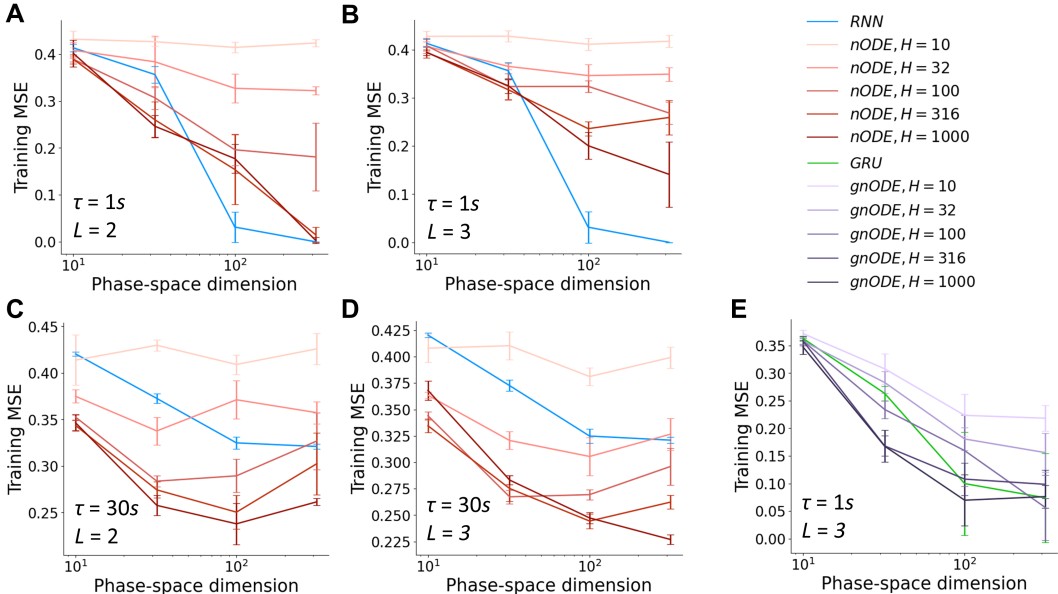

Figure 12: Increasing the number of units $H$ in each hidden layer of $F_\theta$ increases practical expressivity of networks. (A) Training MSEs of nODE with 1 hidden layer (i.e., $L = 2$), assuming $\tau = 1s$. (B) Training MSEs of nODE with 2 hidden layers (i.e., $L = 3$), assuming $\tau = 1s$. (C) Training MSEs of nODE with $L = 2$ and $\tau = 30s$. (D) Training MSEs of nODE with $L = 3$ and $\tau = 30s$. (E) Training MSEs of gnODE with $L = 3$ and $\tau = 1s$.

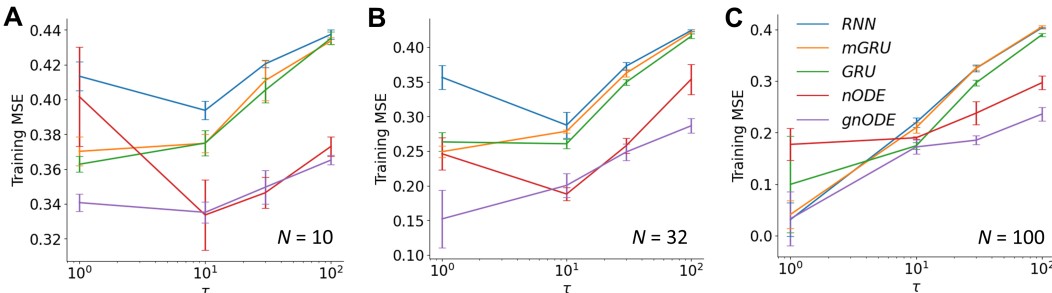

Figure 13: (A) $N = 10$ across all networks. nODE and gnODE has $F_\theta$ with 1 hidden layer, where $H = N_1 = 1000$. (B) $N = 32$ across all networks. nODE and gnODE has $F_\theta$ with 1 hidden layer, where $H = N_1 = 1000$. (C) $N = 100$ across all networks. nODE and gnODE has $F_\theta$ with 1 hidden layer, where $H = N_1 = 1000$.

Lastly, we did not observe that increasing the number of hidden layers in $F_\theta$ significantly increases expressivity of nODE and gnODE (Figure 14). This is perhaps related to the observation that for regression problems, width matters much more than depth (Radhakrishnan et al., 2022).

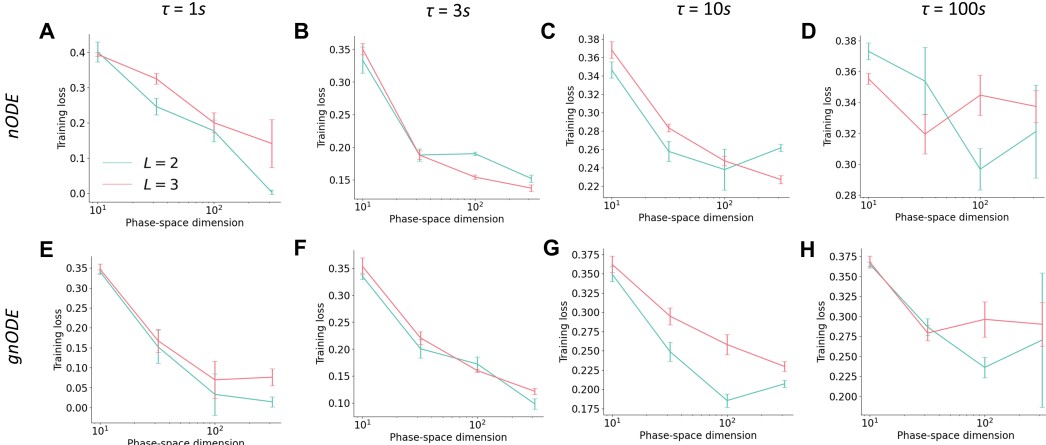

Figure 14: Increasing the number of hidden layers in $F_\theta$ does not significantly increase practical expressivity of networks. (A) nODE with $H = 1000$, assuming $\tau = 1$s. Mint indicates nODE with 1 hidden layer ($H = N_1 = 1000$), and pink indicates nODE with 2 hidden layers ($H = N_1 = N_2 = 1000$). (B) nODE with $H = 1000$, assuming $\tau = 3$s. (C) nODE with $H = 1000$, assuming $\tau = 10$s. (D) nODE with $H = 1000$, assuming $\tau = 100$s. (E) gnODE with $H = 1000$, assuming $\tau = 1$s. (F) gnODE with $H = 1000$, assuming $\tau = 3$s. (G) gnODE with $H = 1000$, assuming $\tau = 10$s. (H) gnODE with $H = 1000$, assuming $\tau = 100$s.

## G    REAL-WORLD TASKS

### G.1    NETWORK INITIALIZATIONS

For RNNs, mGRUs, GRUs, LSTMs and LEMs, we applied either the Kaiming or the Glorot Normal initialization. Both of these initializations should already give criticality. For nODEs and gnODEs, the standard Kaiming or Glorot initialization do not give criticality. Whenever we applied either the Glorot or Kaiming Normal initialization to these networks, the final nonlinearity $\phi_h$ of $F_\theta$ was set to be $\mathcal{I}$. Whenever we applied the critical initialization in Appendix A, $\phi_h$ was set to be tanh. We found that, for all of the real-world datasets we consider (Sections 5.4.1–5.4.3), having no tanh degrades performance, consistent with the observation in (Kidger et al., 2020). We suspect that this is due to the fact that a nODE which does not have does not tanh as its final nonlinearity does not have a chaotic phase (Appendix A).

### G.2 Latin Alphabet Character Trajectory Classification

We trained mGRU, GRU, LSTM, LEM, nODE and gnODE on the task of classifying 20 different Latin alphabets based on the pen-tip trajectories and forces applied to the tip. This dataset had 2858 trials total (2000 trials for training, 429 trials for validation and 429 trials for testing), with each trial being a time-series with 182 time-bins. The time-series was 4-dimensional, with the first dimension being the time stamp, the second being the $x$ position of the pen, the third being the $y$ position, and the fourth being the pen tip force. We took the preprocessed data as is from the Neural CDE repository. Further details on the dataset and the preprocessing step can be found in the repository and Kidger et al. (2020). We used the "30% dropped" dataset (30%, because 30% of the samples in the trajectories were randomly dropped to make the time-series irregularly-sampled) to determine the best set of hyperparameters for each network (using grid search), and used the same set of hyperparameters for the "50% dropped" and "70% dropped" datasets.

We assumed each time-bin in the data is 1s-long (thus each trial is 182s long), and set $\tau = \Delta t = 1$s. We trained our networks for the total of 1300 epochs, where we first trained only the first 14 time-bins for 100 epochs, and then the first 28 time-bins for the next 100 epochs, until we reached 182 time-bins. This method of "iteratively growing the fit" is sometimes used to train a RNN (Hafner, 2017) or a nODE (Rackauckas et al., 2020). The initial states of the networks were also learned – the initial state was assumed to be an affine transformation of the input at the first time-bin. We determined the set of network parameters that achieves the lowest validation loss over the 1300 epochs, and used this validation loss (the cross entropy loss for this classification task) as the measure of performance to determine the best set of hyperparameters. For nODE and gnODE-v1, to determine the best set of hyperparameters, we performed a grid search over the learning rate $\eta \in \{10^{-4}, 10^{-3}, 10^{-2}\}$, rate of weight decay $\lambda_w \in \{10^{-3}, 10^{-2}, 10^{-1}\}$, initialization scheme (Glorot Normal or Kaiming Normal; biases were always initialized with a zero-mean Gaussian with variance $10^{-6}$), phase-space dimension $N \in \{32, 100\}$, and the number of units $H \in \{100, 316, 1000\}$ in each hidden layer of $F_\theta$. For gnODE-v2, we used the same hyperparameter-search space as that of gnODE-v1, except that we applied the critical initialization in Equation 51 of Appendix A. For nODE, gnODE-v1 and gnODE-v2, we only considered FNNs with 2 hidden layers for $F_\theta$ (i.e., $L_h = 3$). For gnODE-v1 and gnODE-v2, we only considered $L_z = 1$. For mGRU and GRU, we performed a grid search over the learning rate $\eta \in \{10^{-4}, 10^{-3}, 10^{-2}\}$, rate of weight decay $\lambda_w \in \{10^{-3}, 10^{-2}, 10^{-1}\}$, initialization scheme (Glorot Normal or Kaiming Normal; biases were always initialized with a zero-mean Gaussian with variance $10^{-6}$) and the phase-space dimension $N \in \{100, 316, 1000\}$. The batch size $B$ was set to be 32 for all networks, following the suggestion in Kidger et al. (2020). With the hyperparameters that achieved the minimum validation loss, we trained the networks 5 times with different random seeds, and evaluated the test losses from those 5 runs. Table 3 shows the best set of hyperparameters found by the grid search, and Table 4 shows the means and standard deviations of each networks' test accuracies for the "30% dropped", "50% dropped" and "70% dropped" datasets with the hyperparameters in Table 3. Table 1A of the main text shows results for the "70% dropped" dataset with the best set of hyperparameters found by the subset of the search space where $N$ is fixed to be 100. The gnODE performance in Table 1A is from gnODE-v2 in Table 4.

We see that the gated networks (mGRU, GRU, LSTM, LEM, gnODE) achieve accuracies similar to that of nCDE reported in Kidger et al. (2020). The gated networks' accuracies are also higher compared to the non-gated network (i.e., nODE). Lastly, we see that mGRU, with fewer parameters, performed similarly to a GRU, confirming that the functional roles of the update and reset gates are similar and the reset gate can be taken out (Krishnamurthy et al., 2022). Similar observations have been made in Ravanelli et al. (2018).

### G.3 Walker2D Kinematic Simulation Prediction

In this experiment, the networks (mGRU, GRU, LSTM, LEM, nODE and gnODE) were given the task of predicting what the future dynamics should be given data samples up until the current time-point in time series generated from the MuJoCo physics engine kinematic simulations (Todorov et al., 2012).

This dataset had $12,893$ trials total (9684 trials for training, 1272 trials for validation and 1937 trials for testing), with each trial being a time-series with 84 time-bins. The time-series was 17-

Table 3: The set of hyperparameters that achieved best performance on the classification of Latin alphabet character trajectories. Along with the best number of parameters identified through the hyperparameter-search, the minimum and maximum number of parameters possible in the search space are also shown in the table. $G$ is Glorot Normal and $K$ is Kaiming Normal initialization. $C$ is the critical initialization proposed in Appendix A. FE = Forward Euler, FBE = Forward-Backward Euler.

| Model | # of parameters (min/max) | $(\eta, \lambda_w, N, N_{\ell_h}, \text{init})$ |
|---|---|---|
| mGRU | $23,520\ (23,520/2,035,020)$ | $(10^{-3}, 10^{-1}, 100, \text{N/A}, G)$ |
| GRU | $312,228\ (34,020/3,040,020)$ | $(10^{-3}, 10^{-3}, 316, \text{N/A}, G)$ |
| LSTM | $4,050,020\ (45,020/4,050,020)$ | $(10^{-3}, 10^{-3}, 1000, \text{N/A}, G)$ |
| LEM (FE) | $415,244\ (45,020/4,050,020)$ | $(10^{-3}, 10^{-3}, 316, \text{N/A}, K)$ |
| LEM (FBE) | $4,050,020\ (45,020/4,050,020)$ | $(10^{-3}, 10^{-3}, 1000, \text{N/A}, G)$ |
| nODE | $33,220\ (17,852/1,208,620)$ | $(10^{-4}, 10^{-3}, 100, 100, K)$ |
| gnODE-v1 | $178,072\ (19,036/1,219,120)$ | $(10^{-3}, 10^{-3}, 100, 316, K)$ |
| gnODE-v2 | $1,219,120\ (19,036/1,219,120)$ | $(10^{-3}, 10^{-3}, 100, 1000, C)$ |

Table 4: Test accuracy for classification of Latin alphabet character trajectories (mean $\pm$ std, error bars computed from training the networks 5 times with different random seeds) with the hyperparameters in Table 3.

| Model | Test Accuracy | | |
|---|---|---|---|
| Name | 30% dropped | 50% dropped | 70% dropped |
| mGRU | $0.987 \pm 0.005$ | $0.987 \pm 0.001$ | $0.983 \pm 0.002$ |
| GRU | $0.990 \pm 0.001$ | $0.990 \pm 0.004$ | $0.987 \pm 0.003$ |
| LSTM | $0.990 \pm 0.002$ | $0.990 \pm 0.004$ | $0.990 \pm 0.002$ |
| LEM (FE) | $0.986 \pm 0.006$ | $0.990 \pm 0.002$ | $0.987 \pm 0.005$ |
| LEM (FBE) | $0.990 \pm 0.004$ | $0.991 \pm 0.004$ | $0.987 \pm 0.001$ |
| nODE | $0.924 \pm 0.095$ | $0.807 \pm 0.247$ | $0.898 \pm 0.089$ |
| gnODE-v1 | $0.984 \pm 0.005$ | $0.981 \pm 0.003$ | $0.986 \pm 0.003$ |
| gnODE-v2 | $0.987 \pm 0.002$ | $0.987 \pm 0.004$ | $0.986 \pm 0.005$ |

dimensional. We took the preprocessed data as is from the ODE-LSTM repository, where $10\%$ of the samples were dropped along the trajectories, and $1\%$ of all actions were overwritten by random actions (Lechner & Hasani, 2020). Further details on the dataset and the preprocessing step can be found in the repository and Lechner & Hasani (2020).

We assumed each time-bin in the data is 1s-long (thus each trial is 84s long), and set $\tau = \Delta t = 1$s. Our networks received the 17-dimensional time-series, along with an extra dimension specifying the time stamp, as input, and emitted their predictions of what the 17-dimensional state will be on the very next time-step. We trained our networks for the total of 700 epochs, where we first trained only the first 14 time-bins for 100 epochs, and then the first 28 time-bins for the next 100 epochs, until we reach 84 time-bins. When we train the full 84 time-bins, we train for 200 epochs instead of 100 epochs, thus making the total 700 epochs. The initial states of the networks were learned – the initial state was assumed to be an affine transformation of the input at the first time-bin. We determined the set of network parameters that achieves the lowest validation loss over the 700 epochs, and used this validation loss (the MSE loss for this prediction task) as the measure of performance to determine the best set of hyperparameters. We performed the same hyperparameter grid search as in Section G.2. The batch size $B$ was set to be 256 for all networks, following the suggestion in Lechner & Hasani (2020). With the hyperparameters that achieved the minimum validation loss, we trained the networks 5 times with different random seeds, and evaluated the test losses from those 5 runs. Table 5 shows the means and standard deviations of each networks' test MSEs along with the hyperparameters found by grid search. The gnODE performance in Table 1B is from gnODE-v2 in Table 5.

Table 5: Test MSE for Walker2D (mean $\pm$ std, error bars computed from training the networks 5 times with different random seeds).

| Model | # of parameters (min/max) | $(\eta, \lambda_w, N, N_{\ell_h}, \text{init})$ | Test MSE |
|---|---|---|---|
| mGRU | $223,113\,(27,417/2,074,017)$ | $(10^{-2}, 10^{-3}, 316, \text{N/A}, G)$ | $1.074 \pm 0.070$ |
| GRU | $328,973\,(39,317/3,093,017)$ | $(10^{-2}, 10^{-3}, 316, \text{N/A}, K)$ | $0.772 \pm 0.028$ |
| LSTM | $53,117\,(53,117/4,131,017)$ | $(10^{-2}, 10^{-3}, 100, \text{N/A}, K)$ | $0.865 \pm 0.009$ |
| LEM (FE) | $440,837\,(53,117/4,131,017)$ | $(10^{-2}, 10^{-3}, 316, \text{N/A}, G)$ | $1.030 \pm 0.022$ |
| LEM (FBE) | $440,837\,(53,117/4,131,017)$ | $(10^{-2}, 10^{-3}, 316, \text{N/A}, G)$ | $0.699 \pm 0.010$ |
| nODE | $1,223,717\,(19,601/1,223,717)$ | $(10^{-3}, 10^{-3}, 100, 1000, G)$ | $0.707 \pm 0.023$ |
| gnODE-v1 | $1,086,833\,(21,233/1,235,617)$ | $(10^{-3}, 10^{-1}, 32, 1000, K)$ | $0.552 \pm 0.019$ |
| gnODE-v2 | $1,235,617\,(21,233/1,235,617)$ | $(10^{-3}, 10^{-3}, 100, 1000, C)$ | $0.588 \pm 0.003$ |

### G.4 SPEECH COMMANDS CLASSIFICATION

We trained mGRU, GRU, LSTM, LEM, nODE and gnODE on the task of classifying ten spoken words, such as "Stop" and "Go", based on one-second audio recordings of these words. The dataset is originally from Warden (2018) and preprocessed using the pipeline in the Neural CDE repository (Kidger et al., 2020). There are total $34,975$ time series (70% training, 15% validation, and 15% test data), where each time series has 20 channels of 161 regularly-sampled data points. Further details on the dataset and the preprocessing step can be found in the repository and Kidger et al. (2020).

Section 5.3 showed that networks perform best when the timescales $\tau$ assumed by the networks match correlation time of the data. To further probe this effect, we varied $\tau$ from $\{0.006s, 0.062s, 0.621s\}$. We trained our networks for the total of 300 epochs on the entire time series. The initial states of the networks were also learned – the initial state was assumed to be an affine transformation of the input at the first time-bin. We determined the set of network parameters that achieves the lowest validation loss over the 300 epochs, and used this validation loss (the cross entropy loss for this classification task) as the measure of performance to determine the best set of hyperparameters. For nODE and gnODE, we tested two different architectures – one without the final tanh nonlinearity (nODE-v1 and gnODE-v1 in Tables 6–8) and one with the tanh (nODE-v1 and gnODE-v2 in Tables 6–8, and gnODE-v3 in Table 6). To determine the best set of hyperparameters, we performed a grid search over the learning rate $\eta \in \{10^{-4}, 10^{-3}, 10^{-2}\}$, rate of weight decay $\lambda_w \in \{10^{-3}, 10^{-2}, 10^{-1}\}$, phase-space dimension $N \in \{100, 316\}$, and the number of units $N_{\ell_h} \in \{1000, 2000, 4000\}$ in each hidden layer of $F_\theta$. We only considered FNNs with 1 hidden layer where $L_h = 2$. For nODE-v1 and gnODE-v1, we applied both the Glorot Normal and Kaiming Normal to determine the best initialization scheme. For nODE-v2, gnODE-v2 and gnODE-v3, we applied the critical initialization (Appendix A). Biases were always initialized with a zero-mean Gaussian with variance $10^{-6}$). For gnODE-v3, the number of units $N_{\ell_z}$ in the hidden layer of $G_\varphi$ was set to be 1000. For mGRU and GRU, we performed a grid search over the learning rate $\eta \in \{10^{-4}, 10^{-3}, 10^{-2}\}$, rate of weight decay $\lambda_w \in \{10^{-3}, 10^{-2}, 10^{-1}\}$, initialization scheme (Glorot Normal or Kaiming Normal; biases were always initialized with a zero-mean Gaussian with variance $10^{-6}$), and the phase-space dimension $N \in \{100, 316, 1000\}$. The batch size $B$ was set to be 256 for all networks. With the hyperparameters that achieved the minimum validation loss, we trained the networks 5 times with different random seeds, and evaluated the test losses from those 5 runs. Tables 6–8 show each networks' test accuracies for $\tau = \{0.006s, 0.062s, 0.621s\}$ with the hyperparameters found by grid-search. The hyperparameter-search space of Tables 7–8 was the same as that of Table 6. The results for nODE and gnODE in Table 1C are from nODE-v2 in Table 6 and gnODE-v3 with $\eta = 0.001$, $\lambda_w = 0.001$, $N = 100$, $N_{\ell_h} = 3500$ and $N_{\ell_z} = 1000$.

Consistent with Figure 3D, we observe that the increased complexity in $F_\theta$ of nODE/gnODE becomes more useful as $\tau$ is increased. When $\tau$ is decreased, the increased complexity in $F_\theta$ does not appear to help. Across all $\tau = \{0.006s, 0.062s, 0.621s\}$, gnODE shows an improvement over nODE, and makes nODE more robust to changes in $\tau$.

When $\tau$ is small (0.006s or 0.062s), nODE/gnODE with the final tanh (i.e., nODE-v2/gnODE-v2) that is critically initialized (following Equation 51 in Appendix A) showed improvement in performance compared to nODE/gnODE without the final tanh (i.e., nODE-v1/gnODE-v1). If there is no

Table 6: Test accuracy (mean $\pm$ std, error bars computed from training the networks 5 times with different random seeds) for classification of Speech Commands with $\tau = 0.006$s.

| Model | # of parameters (min/max) | $(\eta, \lambda_w, N, N_{\ell_h}, \text{init})$ | Test Accuracy |
|---|---|---|---|
| mGRU | $27,610 \ (27,610/2,076,010)$ | $(10^{-2}, 10^{-3}, 100, \text{N/A}, K)$ | $0.809 \pm 0.018$ |
| GRU | $3,098,010 \ (39,810/3,098,010)$ | $(10^{-3}, 10^{-3}, 1000, \text{N/A}, K)$ | $0.855 \pm 0.001$ |
| LSTM | $444,306 \ (54,210/4,142,010)$ | $(10^{-3}, 10^{-3}, 316, \text{N/A}, G)$ | $0.807 \pm 0.004$ |
| LEM (FBE) | $4,142,010 \ (54,210/4,142,010)$ | $(10^{-3}, 10^{-3}, 1000, \text{N/A}, K)$ | $0.836 \pm 0.005$ |
| nODE-v1 | $1,318,438 \ (225,310/2,626,438)$ | $(10^{-4}, 10^{-3}, 316, 2000, G)$ | $0.110 \pm 0.010$ |
| nODE-v2 | $225,310 \ (225,310/2,626,438)$ | $(10^{-3}, 10^{-3}, 100, 1000, C)$ | $0.140 \pm 0.012$ |
| gnODE-v1 | $903,510 \ (237,510/2,733,246)$ | $(10^{-3}, 10^{-3}, 100, 4000, G)$ | $0.795 \pm 0.005$ |
| gnODE-v2 | $771,246 \ (237,510/2,733,246)$ | $(10^{-3}, 10^{-3}, 316, 1000, C)$ | $0.815 \pm 0.004$ |
| gnODE-v3 | $1,972,754 \ (447,410/3,280,754)$ | $(10^{-3}, 10^{-3}, 316, 1000, C)$ | $0.844 \pm 0.002$ |

Table 7: Test accuracy for classification of Speech Commands with $\tau = 0.062$s.

| Model | # of parameters | $(\eta, \lambda_w, N, N_{\ell_h}, \text{init})$ | Test Accuracy |
|---|---|---|---|
| mGRU | $223,738$ | $(10^{-2}, 10^{-3}, 316, \text{N/A}, K)$ | $0.796 \pm 0.002$ |
| GRU | $330,546$ | $(10^{-2}, 10^{-3}, 316, \text{N/A}, K)$ | $0.809 \pm 0.006$ |
| LEM (FBE) | $444,306$ | $(10^{-2}, 10^{-3}, 316, \text{N/A}, K)$ | $0.785 \pm 0.005$ |
| nODE-v1 | $225,310$ | $(10^{-3}, 10^{-3}, 100, 1000, K)$ | $0.246 \pm 0.050$ |
| nODE-v2 | $1,318,438$ | $(10^{-4}, 10^{-3}, 316, 2000, C)$ | $0.169 \pm 0.015$ |
| gnODE-v1 | $1,425,246$ | $(10^{-3}, 10^{-3}, 316, 2000, G)$ | $0.790 \pm 0.018$ |
| gnODE-v2 | $237,510$ | $(10^{-2}, 10^{-3}, 100, 1000, C)$ | $0.769 \pm 0.072$ |

tanh at the end, critical initialization does not appear to help. This may be related to the observation in Appendix A that there is no chaotic regime for networks that only have ReLU activations. This is consistent with the observation in Kidger et al. (2020) that applying tanh at the end improves performance of a nCDE.

For gnODE, using $G_\varphi$ with $L_z = 2$ gave better performance than using $G_\varphi$ with $L_z = 1$ (see gnODE-v2 and -v3 in Table 6).

### G.5  LEM Performance on Real-World Tasks

Our results with LEM on the real-world tasks suggest that the performance of an LEM is similar to an mGRU or a GRU, especially when LEM is discretized with the forward Euler method, instead of the forward-backward Euler method suggested in Rusch et al. (2021) (see Appendix B for a possible explanation and see Appendix G.2, G.3 for LEM results with the forward Euler method). In Rusch et al. (2021), $\Delta t$ in LEM is treated as a hyperparameter, while $\Delta t$ for other models is taken to be equal to 1. For fairer comparisons, $\Delta t$ could have been equal across all models. Our experiment in Section 5.3, and particularly Figure 3D–F makes exactly the point that if we change $\tau$ (which is effectively the same as changing $\Delta t$ in Rusch et al. (2021)), we are effectively giving the models different problems to solve, with different timescales. In our experiments, we set $\tau$ to be the same across all models compared (see also Appendix G.4 for discussion of how changing $\tau$ affects network performances on a real-world task). Rusch & Mishra (2020) discusses training $\Delta t$, and it would be interesting if making $\tau$ trainable in our models leads to improvements in performance, though this would be beyond the scope of this work.

## H  Definition of a Continuous Attractor

We use the terminology "continuous attractor" in the main text, which is very common in neuroscience, but possibly less known in the broader dynamical systems and machine learning communities. In this Appendix, we give a precise definition and attempt to establish a connection between continuous attractors and center manifold theory (Carr, 1981).

Table 8: Test accuracy for classification of Speech Commands with $\tau = 0.621$s.

| Model | # of parameters | $(\eta, \lambda_w, N, N_{\ell_h}, \text{init})$ | Test Accuracy |
|-------|-----------------|--------------------------------------------------|---------------|
| mGRU | $2,076,010$ | $(10^{-2}, 10^{-3}, 1000, \text{N/A}, G)$ | $0.733 \pm 0.005$ |
| GRU | $3,098,010$ | $(10^{-2}, 10^{-3}, 1000, \text{N/A}, G)$ | $0.743 \pm 0.008$ |
| LEM (FBE) | $4,142,010$ | $(10^{-2}, 10^{-3}, 1000, \text{N/A}, K)$ | $0.713 \pm 0.003$ |
| nODE-v1 | $447,310$ | $(10^{-2}, 10^{-3}, 100, 2000, K)$ | $0.710 \pm 0.020$ |
| nODE-v2 | $664,438$ | $(10^{-2}, 10^{-3}, 316, 1000, C)$ | $0.725 \pm 0.009$ |
| gnODE-v1 | $771,246$ | $(10^{-2}, 10^{-3}, 316, 1000, G)$ | $0.720 \pm 0.039$ |
| gnODE-v2 | $771,246$ | $(10^{-2}, 10^{-3}, 316, 1000, C)$ | $0.762 \pm 0.005$ |

By a continuous attractor, we mean a connected manifold of fixed points. More precisely, a first order ODE $\dot{x} = f(x)$ for $x \in \mathbb{R}^n$, is said to have a continuous attractor $S$ if the following conditions hold:

1. $S \subseteq \mathbb{R}^n$ is a $d-$dimensional manifold (usually with a boundary of dimension $d - 1$) embedded in the full phase space, $d \leq n$.
2. $\forall x \in S, f(x) = 0$.
3. Defining the Jacobian $\mathcal{D}(x) = (\partial f / \partial x)(x)$, and the spectral abscissa (or largest real part of the spectrum) $\eta(Df(x))$, then for $x \in S$, the spectral abscissa $\eta(\mathcal{D}(x)) = 0$.

Unlike limit cycles or chaotic attractors, the dynamics is stationary on the continuous attractor $S$, since by definition $\dot{x} = f(x) = 0$ by Item 2. Another almost trivial consequence of the items above are that $S$ is an **invariant manifold** of the dynamics, since for any initial condition $x(0) \in S$, $x(t) \in S$ for all $t \geq 0$. Indeed, $x(t) = x(0)$! Item 3 ensures that perturbations off the manifold $S$ will decay back toward the manifold, implying it is an attractive manifold.

We now want to argue that given Items $1 - 3$ above, $S$ is also a center manifold. Let us now consider the tangent space around a point $x_0 \in S$. This will be spanned by the $d$ zero mode eigenvectors $t_j^k$ of the Jacobian:

$$\sum_{j=1}^{n} \mathcal{D}_{ij} t_j^k = 0, \quad k = 1, ..., d. \tag{66}$$

In other words, we have that $f(x + \epsilon t^k) = 0$ for $k = 1, ..., d$. Let us consider a decomposition of the displacements from $x_0$:

$$x = x_0 + \sum_{k=1}^{d} u_k t^k + \sum_{k=d+1}^{n} y_k n^k. \tag{67}$$

Here we use the fact that nonzero modes $n^k$ will be normal to the manifold. Now we have new global coordinates which align with the tangent space ($u_k$) and transverse space ($y_k$). However, in these coordinates, the constraint for the attractor manifold $S$ becomes

$$f(u, y(u)) = 0. \tag{68}$$

We now seek to determine $y(0)$ and $y'(0)$. By construction, $y(0) = 0$. Taking derivatives of the implicit equation for $S$ gives

$$\frac{\partial f_i(u, y(u))}{\partial u_k} = \sum_j \mathcal{D}_{ij} (\partial x_j / \partial u_k) = \sum_j \mathcal{D}_{ij} \left( t_j^k + \sum_{k'=d+1}^{n} \frac{\partial y_{k'}}{\partial u_k} n_j^{k'} \right) = 0. \tag{69}$$

Since $\mathcal{D} t^k = 0$, this implies

$$\sum_{k'=d+1}^{n} \frac{\partial y_{k'}}{\partial u_k} \sum_j \mathcal{D}_{ij} n_j^{k'} = 0. \tag{70}$$

Since $\mathcal{D}n^k \neq 0$ by construction, we must have that $\partial_{u_k} y_{k'} = 0$, which is what we wanted to show. Therefore, the attractor manifold $S$ is an invariant manifold that is parameterized by a function $y(u)$ which satisfies $y(0) = Dy(0) = 0$. According to (Carr, 1981), this means $S$ is also a center manifold.

## I  COMPUTING INFRASTRUCTURE

We trained our networks on a Dell computer cluster with 320 NVIDIA P100 GPUs across 80 Broadwell nodes. Our networks took up one of these nodes, using one GPU that has 16GB of memory.

