# OpenReview forum: "Gated Neural ODEs: Trainability, Expressivity and Interpretability"
_ICLR.cc/2023/Conference — Submitted to ICLR 2023_

### Official Review · Reviewer_fH84 · 2022-10-24

**Confidence:** 3
**Clarity, Quality, Novelty And Reproducibility:** See comments above.
**Correctness:** 2
**Technical Novelty And Significance:** 2
**Empirical Novelty And Significance:** 2
**Recommendation:** 3

**Strength And Weaknesses:**

## Strength

1) This paper proposes a new model with an interesting combination between the gating mechanism and Neural ODEs.


## Weaknesses

1) The organization of this paper is a bit messy. The paper investigates three largely independent aspects of Neural ODEs with empirical experiments. It is unclear about the synergy between these three aspects. It is also unclear how these aspects relate to "understanding how the dynamics in biological and artificial neural networks implement the computations".

2) Several arguments of the paper are not well-supported. For example, it is uncommon to claim that the model is interpretable just because it can have a relatively lower hidden dimension for achieving the same performance compared to other models, especially when this result is obtained on a single synthetic dataset.

3) The empirical comparison is insufficient. The authors compare the proposed method with very closely relevant baselines such as ODE-LSTM by the reported test MSE in the literature. However, the vanilla Neural ODE and GRU models experimented by this paper already have better test MSE than the reported test MSE of ODE-LSTM in the literature, which casts doubt if the comparison is fair.

4) The paper seems to be a half-baked draft. For example, there is no legend in Figure 3B. The template Author Contributions and Acknowledgements are not removed from the draft.

**Summary Of The Paper:**

This paper proposes a Gated Neural ODE (gnODE) model and empirically investigates the trainability, expressivity, and interoperability of the proposed model.

**Summary Of The Review:**

This is a half-baked draft with significant issues with clarity.

---

> ### Author Response · Authors · 2022-11-19
> **Response to Reviewer fH84**
>
> We thank the reviewer for their constructive feedback of our manuscript. We list our response to each of the reviewer's concerns below:
>
> > The organization of this paper is a bit messy. The paper investigates three largely independent aspects of Neural ODEs with empirical experiments. It is unclear about the synergy between these three aspects. It is also unclear how these aspects relate to "understanding how the dynamics in biological and artificial neural networks implement the computations".
>
> As an instrument for scientific discovery, we believe it is reasonable to ask for a model to be both accurate in fitting the data, and be as simple as possible. Expressivity implies that a model can potentially fit data, whereas trainability implies that  it can fit the data *in practice*. This motivates us studying these two aspects of gnODE. Finally, our take on interpretability is that a dynamical model has a better chance of being interpretable if it is lower-dimensional. We believe this is strikingly illustrated with the variable amplitude flip flop task. But more generally, if there is a choice between accurately modeling with a high-dimensional model vs. a low-dimensional model, it is preferable to choose the lower-dimensional model. Such a model is simpler by the most obvious complexity measure, which is the phase-space dimension. For these reasons, we have chosen to highlight these three aspects in our paper, which we take to be an analysis of a potentially impactful modeling tool, the gated neural ODE.
>
> Regarding the second part of the concern, we have tried to elaborate, in the revised Discussion, how the features of gnODE can be useful in understanding the connection between dynamics and computation in biological and articifial neural networks. While we do not make claims as to whether or not each unit in a gnODE can correspond to a biological neuron, we mention some evidence in the revised Discussion that a biological neural network shows signatures of gating and low-dimensionality, which are the key features of a gnODE that are not present in other models. Several lines of work in neuroscience train RNNs to solve a particular task, and use tools from dynamical systems theory to probe how the RNN implements the solution in its phase-space and compare it to the dynamics of real neural data recorded from animals solving the same task (e.g., [1]). To be able to do this, it is key that one is able to identify the fixed points of the trained networks and analyze the stability of these fixed points as was demonstrated in Figures 1 and 2 of our submission and [1]. With gnODE being trainable and expressive, *while* being more interpretable than other models considered here, we believe it is a good candidate for probing the computations implemented by real neural circuits.

---

> > ### Author Response · Authors · 2022-11-19
> > **Response to Reviewer fH84 (continued-1)**
> >
> > > Several arguments of the paper are not well-supported. For example, it is uncommon to claim that the model is interpretable just because it can have a relatively lower hidden dimension for achieving the same performance compared to other models, especially when this result is obtained on a single synthetic dataset.
> >
> > Previous literature suggests a close relationship between low-dimensionality and interpretability. One often projects high-dimensional data into a low-dimensional projection to understand data better (i.e., via dimensionality reduction). This is done, for example, to understand context-dependent computations in the brain [2, 3]. There is also a line of work which considers low-dimensional dynamics induced by a low-rank structure in the connectivity matrix of an RNN, instead of projecting high-dimensional trajectories onto a lower dimension [4, 5]. This approach has the advantage that, if the rank of perturbations to the connectivity is less than or equal to 3, we can explicitly plot the flow field and tractably analyze the dynamics in low-dimensional space. While these models may be limited in expressivity, the selling point of the gnODE is that it can both be low-dimensional *and* be expressive enough to capture the complexities of a real-world dataset. This is a key feature that makes gnODE stand out as a candidate model for probing the link between computation and dynamics in artificial and biological networks.
> >
> > Regarding the point that the result is obtained on a single synthetic dataset, in fact, we find that gnODE can achieve performance/expressivity similar to other gated RNN models on several carefully designed synthetic datasets and real-world datasets, across different conditions. For synthetic datasets, we find this evidence most notably in Figure 1B and Figure 3D-F. In particular, Figure 3D-F shows that across different $\tau$s and different phase-space dimensions assumed by the models, gnODE is consistently more expressive than other models. In real-world datasets, Table 1 in our revised submission shows that when we fix the phase-space dimension to be reasonably low (in the main text we fix $N=100$, and explore other possibilities in Appendix G), gnODE either outperforms or at least performs on par with other models considered in this work on all three real-world datasets.

---

> > > ### Author Response · Authors · 2022-11-19
> > > **Response to Reviewer fH84 (continued-2)**
> > >
> > > > The empirical comparison is insufficient. The authors compare the proposed method with very closely relevant baselines such as ODE-LSTM by the reported test MSE in the literature. However, the vanilla Neural ODE and GRU models experimented by this paper already have better test MSE than the reported test MSE of ODE-LSTM in the literature, which casts doubt if the comparison is fair.
> > >
> > > We thank the reviewer for bringing up concerns about the fairness of comparison. As the reviewer correctly points out, the test MSEs of vanilla nODE and GRU models in this paper is lower than the test MSEs of ODE-LSTM in [6]. This is because, as we discuss in the first paragraph of Section 5.5 of the original submission (now Section 5.4.2 of the revised submission), we use natural cubic splines to interpolate missing data. The models compared in [6] do not use sophisticated interpolating schemes. It is generally known that applying appropriate interpolating schemes for irregularly-sampled time-series datasets can be critical to a model's performance [7] and is also a part of a model [8, 9]. We applied the same interpolating scheme (natural cubic spline) across the network architectures considered in this paper because this scheme seemed to make more sense for the prediction of kinematic dynamics. The comparisons in [6] do not assume same interpolation scheme across models (e.g., CT-RNN/GRU/LSTMs use an interpolation scheme, but GRU-D/Bi-directionRNN/ODE-RNN/ODE-LSTM and others do not). Our results in Table 1 (and Appendix G.3) suggest that, if the focus is entirely on which model achieves better performance, clearly gnODE with natural cubic spline is better than other models. If the focus is in identifying whether gating helps nODEs to perform better (controlling for interpolating schemes), then gating does seem to boost performance of a nODE in all of the datasets we tested. If the focus is on what interpolation scheme helps improve performance on this dataset, then it does appear to be that interpolating with natural cubic splines is better than the interpolation scheme used in [6] and better than not using an interpolation scheme at all. Therefore, in sum, we believe the comparisons made in the manuscript were fully fair. Other than interpolation schemes, we also tried to be careful about the choice of discretization and used the same discretization scheme across all architectures to make sure that the improvements we see is not due to using a different discretization/interpolation schemes but because of the architecture, particularly gating.
> > >
> > > To further address the reviewer's concern, in the revised submission, we run additional experiments on:
> > >
> > > * a new, fairly complex task of classifying ten spoken words from audio recordings of these words
> > >
> > > * comparisons with the LSTM on all real-world datasets
> > >
> > > * different discretizations (forward Euler and forward-backward Euler) of the LEM (to ensure fairer comparisons)
> > >
> > > * whether/how critical initialization of nODE/gnODE determined in Appendix A of our original submission helps with trainability
> > >
> > > * whether/how using a feedforward neural network for the gating function $G_\varphi$ helps performance
> > >
> > > * whether/how the intrinsic $\tau$ of the network affects training (as was originally predicted by Figure 3 of the manuscript)
> > >
> > > Section 5.4 and Appendix G in the revised submissions reflect these changes.

---

> > > > ### Author Response · Authors · 2022-11-19
> > > > **Response to Reviewer fH84 (continued-3)**
> > > >
> > > > > The paper seems to be a half-baked draft. For example, there is no legend in Figure 3B. The template Author Contributions and Acknowledgements are not removed from the draft.
> > > >
> > > > We respectfully disagree with the reviewer that the submission should be considered a half-baked draft based on cosmetic issues such as uncommented template blurbs for Author Contributions and Acknowledgements.
> > > >
> > > > Regarding the claim that Figure 3B has no legend, we would like to reiterate that the color scheme for Figure 3B is the same as Figure 3A. We mention in the caption that (B) has training MSEs of gnODEs in (A) in the original submission. Similarly, (E-F) do not have legends because they use the same color scheme as in (D).
> > > >
> > > > We removed the Author Contributions and Acknowledgements from the revised anonymous version of the draft to address the reviewer's point.
> > > >
> > > > [1] Flexible multitask computation in recurrent networks utilizes shared dynamical motifs. Driscoll et al., 2022.
> > > >
> > > > [2] Context-dependent computation by recurrent dynamics in prefrontal cortex. Nature. Mante et al., 2013.
> > > >
> > > > [3] Neural population dynamics during reaching. Nature. Churchland et al., 2012.
> > > >
> > > > [4] Linking connectivity, dynamics, and computations in low-rank recurrent neural networks. Neuron. Mastrogiuseppe and Ostojic, 2018.
> > > >
> > > > [5] Parametric control of flexible timing through low-dimensional neural manifolds. Beiran et al., 2021.
> > > >
> > > > [6] Learning Long-Term Dependencies in Irregularly-Sampled Time Series. Lechner \& Hasani, 2020.
> > > >
> > > > [7] Neural Controlled Differential Equations for Online Prediction Tasks, Morrill et al., 2021
> > > >
> > > > [8] Neural Controlled Differential Equations for Irregular Time Series. Kidger et al., NeurIPS, 2020.
> > > >
> > > > [9] Neural Rough Differential Equations for Long Time Series. Morrill et al., 2021

---

### Official Review · Reviewer_oDgk · 2022-10-24

**Confidence:** 4
**Correctness:** 3
**Technical Novelty And Significance:** 3
**Empirical Novelty And Significance:** 3
**Recommendation:** 8

**Clarity, Quality, Novelty And Reproducibility:**

Nicely written paper. I am unsure about the interpretability of section with D=2 given that for most applications these nODEs will work in larger dimensions. How can you translate it to higher dimensions is unclear to me?



**Strength And Weaknesses:**

What the authors call the principle of expressivity is a very interesting phenomena of doing more with less. That reveals itself best fitting random trajectories indicates that gnODEs can potentially thread through unstable saddles as shown in Fig. 3

The simple proposal of using the dissipative -h term to achieve stability for many initial conditions might be perceived as trivial for people working in dynamical systems. And it's surprising that other have not used before.

It’d be informative to also plot the time scaling of the vector field in each of the tasks. Is gnODE freezing the dynamics when learning the variable amplitude flip-flop?

Can the authors elaborate on the structure of the marginally-stable fixed points for a given autonomous dynamical system realization? The presence of unstable saddle is interesting because those are the main ingredients of heteroclinic orbits which are structurally unstable but can create sequences of quasi-stable states that are input dependent. This type of dynamics is referred in the literature as winnerless-competition principle.

For a given size N, gNN and non-time dependent input, how many stable and saddle fixed points are detected? GRU and mGRU appear to learn only stable fixed points which is not an interesting dynamics. gnODE, on the other hand, appear to have a combination of saddles and stable FPs, but I’d like to know if the system has a complex structure of fixed points for non-explicit time dependence on the inputs (autonomous dynamical system). Then, tracking the separatrices of the saddles would enrich our understanding of what gnODE can learn.

Can the authors explain what they mean by continuous attractors? There most common attractors one finds in these types of dissipative dynamical systems are limit cycles or even chaotic attractors. Are the authors referring to center manifolds?


**Summary Of The Paper:**

This paper advances the applicability and interpretability of neural ODE by injecting a regulatory mechanism to adjust the conditional time scale of the vector field. The authors show improvement on a variety of tasks respect to previous methods, and provide insights about the structure of the fixed points by inspecting the maximum eigenvalue of the Jacobian.

**Summary Of The Review:**

Intriguing paper that I enjoy reading. It shows better performance in a variety of tasks especially when the nODE needs to learn complex random trajectories. Yet, many questions remain in terms of what's the underlying dynamical structure generated in the autonomous regime.

---

> ### Author Response · Authors · 2022-11-19
> **Response to Reviewer oDgk**
>
> We thank the reviewer for their positive and constructive feedback. We list the responses to each of the reviewer's points below:
>
> > Can the authors elaborate on the structure of the marginally-stable fixed points for a given autonomous dynamical system realization? The presence of unstable saddle is interesting because those are the main ingredients of heteroclinic orbits which are structurally unstable but can create sequences of quasi-stable states that are input dependent. This type of dynamics is referred in the literature as winnerless-competition principle.
>
> The marginally-stable fixed points are observed for the variable amplitude flip-flop task. This task *can* be solved by integrating the inputs, and we observe that the networks we train to solve it appear to pick up this functionality. In other words, they solve it precisely by integrating the inputs. Given this functionality, we believe, and therefore argue, that the marginally-stable fixed points are used to integrate inputs. In order to accomplish this, they must be highly structured in phase space, and form what we refer to as an approximate continuous attractor (defined below). The simplest example would be a number of marginally-stable fixed points arranged on a line, and inputs pushing the state in one or the other direction along this line, from one marginal FP to the next.
>
>
>
>
> > For a given size N, gNN and non-time dependent input, how many stable and saddle fixed points are detected? GRU and mGRU appear to learn only stable fixed points which is not an interesting dynamics. gnODE, on the other hand, appear to have a combination of saddles and stable FPs, but I’d like to know if the system has a complex structure of fixed points for non-explicit time dependence on the inputs (autonomous dynamical system). Then, tracking the separatrices of the saddles would enrich our understanding of what gnODE can learn.
>
> This is a very interesting question. We agree that understanding the phase space structure of the gnODE would certainly enrich our understanding of its expressivity, and thank the reviewer for this suggestion. We indeed studied the structure of the fixed points in the variable amplitude flip-flop task, but chose instead to show the flow vector field instead, as we felt it was more illustrative of the phenomenon we were trying to demonstrate (integrator functionality and attractor structure). In Appendix E.4, we discuss our analysis of the structure and distribution of fixed-points for the vanilla RNN in this setting. We projected the 100-dimensional fixed points to the 3-dimensional PC space and found that the unstable fixed points are scattered around the stable fixed points, suggesting that the unstable fixed points may be facilitating the network to fall into one of the stable or marginally stable fixed points.
>
> > Can the authors explain what they mean by continuous attractors? There most common attractors one finds in these types of dissipative dynamical systems are limit cycles or even chaotic attractors. Are the authors referring to center manifolds?
>
>
> In our manuscript, we use the terminology "continuous attractor", which is very common in neuroscience, but possibly unknown in the broader dynamical systems and machine learning communities. We thank the reviewer for this opportunity to clarify and attempt to bridge the language used by both communities. In the manuscript, we have included a reference to a review [1] which defines continuous attractors and discusses their role in neuroscience. For convenience, we review the definition here, and comment on its connection to center manifold theory. By a continuous attractor, we mean a connected manifold of fixed points. More precisely, a first order ODE $\dot{x} = f(x)$ for $x \in \mathbb{R}^{n}$, is said to have a continuous attractor $S$ if the following conditions hold:
>
> 1. $S \subset \mathbb{R}^{n}$ is a $d$-dimensional manifold (usually with a boundary of dimension $d-1$) embedded in the full phase space, $d < n$.
>
> 2.  $f(x) = 0$  $\forall x \in S$.
>
> 3. Defining the Jacobian $\mathcal{D}(x) = \frac{\partial f}{\partial x}(x)$, and the spectral abscissa (or largest real part of the spectrum) $\eta(Df(x))$, then for $x\in S$, the spectral abscissa $\eta(\mathcal{D}(x))  = 0$.
>
> Unlike limit cycles or chaotic attractors, the dynamics is stationary on the continuous attractor $S$, since by definition $\dot{x} = f(x) = 0$ by Item (2). Another almost trivial consequence of the items above are that $S$ is an **invariant manifold** of the dynamics, since for any initial condition $x(0) \in S$, $x(t) \in S$ for all $t\ge 0$. Indeed, $x(t) = x(0)$!
>
> We have added a new Appendix H which includes the definitions above, as well as a short proof that the continuous attractor is also a center manifold for the points in $S$.

---

> > ### Author Response · Authors · 2022-11-19
> > **Response to Reviewer oDgk (continued)**
> >
> > > I am unsure about the interpretability of section with D=2 given that for most applications these nODEs will work in larger dimensions. How can you translate it to higher dimensions is unclear to me?
> >
> > Many applications of gnODEs may have $D>2$. However, our results suggest that gnODEs with lower $D$s generally can perform competitively against other recurrent networks with higher $D$s. If there was a choice between accurately modeling with a high-dimensional model vs. a low-dimensional model, it would be preferable to choose the lower-dimensional model.
> >
> > > It’d be informative to also plot the time scaling of the vector field in each of the tasks. Is gnODE freezing the dynamics when learning the variable amplitude flip-flop?
> >
> > We take the question to be of the following: For the part of the phase space that gives $\dot{\boldsymbol{h}} = \boldsymbol{0}$, is it the gating component (i.e., $\sigma(\boldsymbol{W}^0_z \boldsymbol{h} + \boldsymbol{U}_z \boldsymbol{x} + \boldsymbol{b}^0_z)$) that is outputting $\boldsymbol{0}$ or the other side (i.e., $-\boldsymbol{h} + F_\theta(\boldsymbol{h},\boldsymbol{x})$)? When we evaluated the gate output and $-\boldsymbol{h} + F_\theta(\boldsymbol{h},\boldsymbol{x})$ on several points inside the 2-D continuous attractor, the norm of $-\boldsymbol{h} + F_\theta(\boldsymbol{x},\boldsymbol{h})$ was closer to 0 than the norm of the gate output, but the gates still yielded small values close to ~0.1. We interpret this to mean that the network learns a very slow region by utilizing a combination of structural fixed points (from $-\boldsymbol{h} + F_\theta(\boldsymbol{x},\boldsymbol{h}) \approx \boldsymbol{0}$) with long effective time constants (due to a small value of the gating variable).
> >
> > [1] Computational principles of memory. Chaudhuri and Fiete, Nature Neurosci., 2016
> >
> > [2] Applications of Centre Manifold Theory. Carr, 1981

---

### Official Review · Reviewer_XxBD · 2022-10-24

**Confidence:** 4
**Correctness:** 3
**Technical Novelty And Significance:** 3
**Empirical Novelty And Significance:** 3
**Recommendation:** 6

**Clarity, Quality, Novelty And Reproducibility:**

The manuscript is clear and high quality, well written and comprehensive description on the properties of a novel neural ODEs architecture.

**Strength And Weaknesses:**

The paper is comprehensive description of the gated neural ODE and provides  basic intuition of its capabilities. There was one practical example on its capability to fit continuous processes. The strength of the paper is that is describes a novel model that could "explain" or "interpret "different data sets and provide a model that has the ability to shed light  to the fundaments in the process described by the data, and, for example, show a fixed point structure in the particular data.

The weak point is that this remains quite an exercise of experimental trainings on selected examples without practical use cases. The question that is hinted at the introduction remains open. Could this be a tool that can reveal some of the algorithms biological neural networks are running.

**Summary Of The Paper:**

The manuscript aims to build an accurate, explainable digital twin for the biological neural networks. The neural ODEs have a capability to address complex behaviour. The Authors suggest a gated version to ensure also long term memory effects. The gated systems, the authors find are leading to simpler dynamics, all in increase the trainability, expressivity and interpretability. The dynamics of the gated neural ODE is discussed quite lengthly relating to the other already existing alternatives.

Experimental trials are made for N-bit flip flop task, fitting a finite number of samples from an Ornstein-Uhlenbeck (OU) process, and a 2d walker kinematic simulation predictor among others. The experiments shows that the gated version can utilise lower dimensional phase spaces. The paper provides an extensive Appendix where the performance of the gated system is studied from multiple point of views.


**Summary Of The Review:**

Excellent, experimental review the performance of gated neural ODEs with some mathematical analysis in the appendices.  A good addition to the toolbox for creating models for multi-scale signals.  The capability to the architecture is demonstrated with a wide set of examples. However, I am waiting for the real process with importance that can be "understood" and "interpreted" by using this architecture in the fit. For example, a discovery of type of biological neural system that can be characterised and interpreted with this novel tool.

---

> ### Author Response · Authors · 2022-11-19
> **Response to Reviewer XxBD**
>
> We thank the reviewer for the constructive feedback of our manuscript. We list our response to the reviewer’s concerns below:
>
> > The weak point is that this remains quite an exercise of experimental trainings on selected examples without practical use cases.
>
> We thank the reviewer for pointing this out. While Sections 5.2 and 5.3 are carefully designed synthetic tasks to get an intuition on how gnODE works compared to other models, Sections 5.4 and 5.5 in the original submission were concerned with how practical it is to use gnODE on some real-world tasks that involve sequences. However, given that there were only two practical tasks in the original submission, in the revised submission, we newly added a third, fairly complex, real-world task of classifying ten spoken words from audio recordings of these words. In the revised submission, a new Section 5.4 called "Real-World Tasks" compares gnODE to other models on three different real-world tasks.
>
> > The question that is hinted at the introduction remains open. Could this be a tool that can reveal some of the algorithms biological neural networks are running.
>
> We streamlined the Discussion of our revised submission, so that there is a discussion on the biological relevance of gnODE. As was done in [1, 2] and similarly in Figure 2 of our original and revised submission, we can compute the spectral abscissa of the Jacobian around the fixed points to characterize how the network dynamics implement the algorithms for solving a certain computational task.
>
> > However, I am waiting for the real process with importance that can be "understood" and "interpreted" by using this architecture in the fit. For example, a discovery of type of biological neural system that can be characterised and interpreted with this novel tool.
>
> We thank the reviewer for this point. Previous literature suggests that a nODE architecture is useful for inferring the low-dimensional latent dynamics underlying neural population activity [3]. While we do not attempt to infer latent neural population dynamics in this work, given our experiments suggesting that gnODE improves the trainability, expressivity and interpretability of a nODE, we expect that a gnODE-based architecture will be even more powerful than a nODE-based architecture used in [3] in inferring the latent neural dynamics in a real brain.
>
> [1] Opening the black box: low-dimensional dynamics in high-dimensional recurrent neural networks. Neural Comput. Sussillo \& Barak, 2013.
>
> [2] Flexible multitask computation in recurrent networks utilizes shared dynamical motifs. Driscoll et al., 2022.
>
> [3] Inferring latent dynamics underlying neural population activity via neural differential equations. ICML. Kim et al., 2021.

---

### Official Review · Reviewer_5GnR · 2022-10-26

**Confidence:** 4
**Correctness:** 3
**Technical Novelty And Significance:** 3
**Empirical Novelty And Significance:** 3
**Recommendation:** 5

**Clarity, Quality, Novelty And Reproducibility:**

The  paper is generally clearly written, the methods are described clearly. The motivation for the work could be explained a bit more.

The work is novel to my knowledge.

Based on the given experimental details, it does look like the work can be reproduced.

**Strength And Weaknesses:**

## Strengths:

- The method is a simple extension of neural ODEs that seem to have many useful properties and benefits.
- The ability to learn continuous attractors in an interpretable way is interesting.
- Performs well on considered benchmarks.
- The expressivity measure seems useful and relevant to the community.

## Weaknesses:

- Very limited set of benchmarks are considered.
- No comparison with architectures such as LSTMs, LMU, S4 etc.
- The specific motivation of constructing this architecture is not clear -- is it to provide a better performing architecture? If so comparison on lots more benchmarks and architectures are needed.
- gnODE seems to be very parameter inefficient compared to other methods here.

## Other questions:

- Fig. 1: why does gnODE have much higher variance than other RNN models?

**Summary Of The Paper:**

The authors introduce a gated neural ODE, an architecture that extends the neural ODE with a recurrent gating variable. They demonstrate that this architecture can learn continuous attractors, provide somewhat interpretable solutions and perform well for some select benchmarks.

**Summary Of The Review:**

Overall, the gnODE is a very interesting architecture that might have the potential to be an interpretable performant architecture. But the paper only uses a narrow set of benchmarks and compares with very few other relevant architectures, so the generality and scalability of the model proposed here is not clear.

---

> ### Author Response · Authors · 2022-11-19
> **Response to Reviewer 5GnR**
>
> We thank the reviewer for the constructive feedback. We have tried our best to incorporate the suggested changes, and we feel this has improved the revised submission. We list our response to each of the reviewer's concerns below:
>
> > Very limited set of benchmarks are considered. No comparison with architectures such as LSTMs, LMU, S4 etc...  the paper only uses a narrow set of benchmarks and compares with very few other relevant architectures, so the generality and scalability of the model proposed here is not clear.
>
> We thank the reviewer for encouraging us to test our model (gnODE) against more architectures and on more benchmarks. We made major revisions to Sections 5.4 and 5.5 in the original submission, so that the real-world tasks are now under the new Section 5.4 (Real-World Tasks). Additionally, we include a new, fairly complex task of classifying ten spoken words from audio recordings of these words (Section 5.4.3 Speech Commands Classification). Furthermore, as a comparison, we include results from LSTM on all of the benchmarks. We also add an extensive appendix (Appendix G) that provides details on the experiments and a variety of nODE and gnODE architectures that we tested, along with a discussion on how LEM with different discretizations (Forward Euler and Forward-Backward Euler) performs on our benchmarks.
>
> One aspect we newly explore in the revised submission is whether or not the critical initialization of nODE/gnODE determined in Appendix A helps with nODE/gnODE performances on the benchmarks. We find that
> when critical initialization is applied to these networks without the final tanh nonlinearity to $F_\theta$, the performance is worse than other popular initializations (e.g., Glorot or Kaiming normal). However, when critical initialization is applied with the final tanh nonlinearity, we find improvements in performance in some of the benchmarks. We suspect that having the tanh nonlinearity is helpful because, as observed in Appendix A, without the final tanh, the networks do not have chaotic regime, which means the autonomous dynamics is unstable and the latent-state grows without bound.
>
> Lastly, we added results on the effect of making the gating function more expressive. Specifically, instead of the ``single layer'' gating function $G_\varphi (\boldsymbol{h}, \boldsymbol{x})= \sigma(\boldsymbol{W}^0_z \boldsymbol{h} + \boldsymbol{U}_z \boldsymbol{x} + \boldsymbol{b}^0_z)$, we used a feedforward neural network with a sigmoid nonlinearity in the end. As we show in Appendix G.4, this improves performance.
>
> In summary, in the revised submission, we added the following items to address the reviewer's concerns:
>
> * a new benchmark (Speech Commands)
>
> * comparison with the LSTM on all benchmarks
>
> * different discretizations of the LEM (to ensure fairer comparisons)
>
> * whether/how critical initialization of nODE/gnODE helps training
>
> * whether/how using a more expressive, feedforward neural network for the gating function $G_\varphi$ helps
>
> We hope that these revisions address the reviewer's concern.

---

> > ### Author Response · Authors · 2022-11-19
> > **Response to Reviewer 5GnR (continued)**
> >
> > > gnODE seems to be very parameter inefficient compared to other methods here.
> >
> > We thank the reviewer for bringing up this point. We explain below why comparing parameter counts between models is tricky.
> >
> > In our original submission, we did a hyperparameter grid search over the initialization schemes (Glorot and Kaiming normal), learning rate (and decay rate of ADAMW), the phase-space dimension and the number of units in the hidden layer of $F_\theta$ (for nODE and gnODE). We then reported in Table 1 of the original submission the test accuracies and MSEs of the models that achieved the best performances in the validation datasets.
> >
> > However, because the total number of parameters varied by large amounts across the hyperparameter-search space, it is hard to say that one model is more parameter-efficient than the other, just by looking at the number of parameters of a model with the set of hyperparameters that achieved best performance on the validation dataset. Therefore, given that the point that we are trying to make is not whether or not a model is more parameter-efficient than the other, we decided to move this information to Appendix G (now along with the minimum and maximum possible number of parameters possible in the hyperparameter-search space). For example, for Walker2D, this information is now in Table 5 of Appendix G.3 in our revised manuscript.
> >
> > Table 5 shows that even a GRU with the number of parameters as large as ~3 million failed to achieve a better performance compared to a gnODE with ~1 million parameters on the Walker2D task. This is consistent with the previous literature on nODEs, that whenever the intrinsic dimenionality of the task is low (which appears to be the case for Walker2D, as the phase-space dimension could be as low as $32$ and gnODE could still perform quite well), nODEs appear to be more parameter efficient than RNNs.
> >
> > In the main text, we are now fixing the phase-space dimension $N$ to be $100$, and reporting the test accuracies and MSEs of different models for the three benchmarks, similar to Section 5.2. We find that gnODE performs competitively against other architectures considered in these benchmarks.
> >
> > > The specific motivation of constructing this architecture is not clear -- is it to provide a better performing architecture? If so comparison on lots more benchmarks and architectures are needed... The motivation for the work could be explained a bit more.
> >
> >
> > The motivation is described in the second paragraph of Section 3. We repeat the text here for convenience:
> >
> > (From Section 3) "The primary motivation for introducing gating is its robust ability to generate long timescales and to address the exploding and vanishing gradients problem (EVGP). Our work can be viewed as distilling the key elements of gating from GRUs and LSTMs responsible for long timescales and stable gradients, and incorporating them in nCDE [or nODE]-inspired models."
> >
> > We further show in Appendix B an analysis of the empirical Jacobian spectrum of the architectures we consider in the main text. We show that the gated architectures, including gnODE, show "a robust pinching, leading to eigenvalues clustering near zero and thus long timescales/stable gradients... This pinching results in long-lived modes, contributing to all of these gated networks' ability to learn long time dependencies".
> >
> > Another motivation for introducing the gnODE is that in addition to generating long timescales, the gate can also give rise to *adaptive* timescales which adjust depending on the inputs. We now explore aspects of this question in Appendix G.4 of the revised submission by varying the intrinsic time-constants - $\tau$ - of the different models, and we find that while non-gated models (e.g., nODE) are sensitive to finding the right $\tau$ for the task, the gnODE is much more robust to this, likely due to the adaptive timescale afforded by the gate.
> >
> > > Fig. 1: why does gnODE have much higher variance than other RNN models?
> >
> > We thank the reviewer for this point. In this regard, we would like to point out that the same hyperparameter grid search was used for all the models. It is likely that the range of hyperparameters for which the different models train well could be different, thus making a raw comparison of the variance not very informative. Specifically, the 27 points in Figure 1B and D are different hyperparameter configurations where the learning rate (of ADAMW) was selected from {$10^{-2}$, $10^{-3}$, $10^{-4}$}, the decay rate (of ADAMW) was selected from { $10^{-1}$, $10^{-2}$, $10^{-3}$ }, and the batch size was selected from {$10$, $50$, $100$}. From our experiments with the real-world tasks, the higher variance does not appear to be due to some property of nODE/gnODE (as shown in the error bars of Table 1 in the main text and Tables 4-8 in the Appendix), and we do not make any claims regarding variance in our manuscript.

---

> > > ### Comment · Reviewer_5GnR · 2022-12-12
> > > **Thank you for the response**
> > >
> > > I thank the authors for the response, which has addressed some of my concerns with the addition of the new benchmark and LSTM comparison. But I will maintain my original score for the following primary reason:
> > > It is mentioned that the motivation of the paper is to "generate long time scales". But this phrase is still unclear for me -- I interpret this to mean that you want to model long range dependencies? If so, the empirical evidence is quite insufficient because of lack of comparison with key recent models that also address the same issue (LMU, S4, etc.) as I mentioned in my original comment.
> > >
> > > Apart from that, I also find the discussion of the parameter efficiency confusing, since on one hand you want to show that these models are expressive using a measure dependent on the number of parameters, and yet the hyper-parameter search yields models with large number of parameters for the same tasks. Moreover, I usually expect performance comparisons to be done with models having comparable number of parameters/computations.
> > >
> > > Overall, I think there is a gap between what the authors want to address and what they are addressing in this paper. But I think the ability of the model to learn continuous attractors that can potentially lead to interpretable models very interesting. With a more rigorous empirical study and more focus, the paper can become a very valuable and relevant for the community.

---

> > > > ### Author Response · Authors · 2022-12-12
> > > > **Response to Reviewer 5GnR**
> > > >
> > > > We thank the reviewer for the reply. We would like to mention that even though we did not compare our model to S4 and LMU, we did demonstrate the benefits of gating for previously non-gated architectures. Moreover, we did a thorough comparison with models based on gating mechanisms, including the recent LEM model, the GRU, LSTM and mGRU.
> > > >
> > > > We believe that while our empirical results do not show that gnODE is state-of-the-art, they do show that gating not only helps RNNs achieve better performance, but also more general architectures such as FNNs in vanilla nODEs, which can be low-dimensional and interpretable. We would also like to further clarify the reviewer's point on parameter efficiency. We do in fact find that gnODEs with *lower* number of parameters perform as well as or better than other models tested, except for the Speech Commands dataset on the low $\tau$ condition (as shown in Appendix G). Thus, we respectfully disagree with the claim that our empirical results on real datasets are inconsistent with our results on expressivity.

---

> > > > > ### Comment · Reviewer_5GnR · 2022-12-12
> > > > > **parameter efficiency**
> > > > >
> > > > > > We do in fact find that gnODEs with lower number of parameters perform as well as or better than other models tested
> > > > >
> > > > > I see other models with fewer parameters do better in almost all the tasks (maybe I'm misundestanding something?). For example in Table 3/4, mGRU with 23K parameter does better than gnODE-v1/v2 with 178K/1.2M parameters. In Table 5, all the other listed models have fewer parameters. The other models also seem to have higher MSEs, but did increasing their parameters decrease MSE? Otherwise it's a bit of an unfair comparison. In Table 8, nODE-v1/v2 have fewer parameters than gnODE-v1/v2. Which comes back to my point that the empirical evaluation is not robust enough -- comparing all models with similar number of parameters would have helped make this point a lot clearer. As it is, the best performance has been reported, but the performance is not always comparable either. So the overall message is not clear, at least to me.

---

> > > > > > ### Author Response · Authors · 2022-12-12
> > > > > > **Re: parameter efficiency**
> > > > > >
> > > > > > Thank you for pointing this out. However, we do not have the same reading of the results. Specifically,
> > > > > >
> > > > > > > For example in Table 3/4, mGRU with 23K parameter does better than gnODE-v1/v2 with 178K/1.2M parameters.
> > > > > >
> > > > > > Our reading of Table 3/4 is that mGRU with 23K parameters (and mGRU that is even as large as 2M) performed similarly to (not better than) gnODE-v1/v2 with 178K/1.2M.
> > > > > >
> > > > > > > In Table 5, all the other listed models have fewer parameters. The other models also seem to have higher MSEs, but did increasing their parameters decrease MSE?
> > > > > >
> > > > > > Increasing their parameters increased MSE on the validation dataset. An LEM or LSTM as large as 4M could not perform as well as gnODE with 1M parameters.
> > > > > >
> > > > > >
> > > > > > > In Table 8, nODE-v1/v2 have fewer parameters than gnODE-v1/v2.
> > > > > >
> > > > > > Similarly, nODE as large as 2M could not perform as well as gnODE with 1M or less.
> > > > > >
> > > > > > We agree that we could have shown a slice of the results of the HP search, fixing the number of parameters across models. Since a selling point of gnODE is that we can constrain/fix the phase-space dimension and achieve similar/better performance than other models (as suggested by Figure 1 and 3), we showed results with the phase-space dimension fixed same across models in the Table of the main text. We hope that this clarifies some of your concerns.

---

### Author Response · Authors · 2022-12-08
**Note to the Reviewers and the Area Chair**

Dear Reviewers and Area Chair,

Thank you again for your efforts in reviewing our paper. With the discussion period coming close to an end, we kindly ask if you have feedback or thoughts on our rebuttal. We have tried our best to address each of the reviewers concerns and believe that this has significantly improved the manuscript. We would be more than happy to answer any questions or discuss any further concerns until the deadline. We hope that our rebuttal addresses your concerns and hope you consider our rebuttal in your final assessment.

Best Regards,

Authors

---

### Decision · Program_Chairs · 2023-01-20

**Decision:**

Reject

**Justification For Why Not Higher Score:**

While the work received at least partial support from some of the reviewers, the significance of the paper appears relatively limited.

**Justification For Why Not Lower Score:**

NA

**Metareview: Summary, Strengths And Weaknesses:**

This paper introduces an extension to neural ODEs to include a recurrent gating variable. The architecture is shown to be able to learn continuous attractors, while yielding somewhat interpretable solutions and performing relatively well on some benchmarks. There was general consensus among the reviewers that the interpretability results were interesting. However, concerns were raised regarding the  limited and mostly synthetic benchmarks without any obvious practical use cases.

**Summary Of Ac-Reviewer Meeting:**

We had an AC-reviewer meeting but unfortunately two of the reviewers never responded to my requests for scheduling the meeting. Hence only 2 of the reviewers attended the meeting. The 2 reviewers were in general agreement that the intepretability experiments were interesting but that the significance of the work was quite limited beyond this. The reviewers also discussed closely related work by Collins et al. (Capacity and Trainability in Recurrent Neural Networks. ICLR 2017) and agreed that the analyses presented here were directly borrowed from that work (and that the paper should have been cited). Experimental limitations were also discussed in that architectures with potentially very different numbers of parameters and/or units were compared and no effort was attempted to control for these -- which is quite problematic for a study on the capacity of these network architectures.